# Ligand-mediate exciton allocation enables efficient cluster-based white light-emitting diodes via single and heavy doping

Jianan Sun[1], Naiyu Li[1], Zhuke Gong[1], Yi Man[1], Chunlei Zhong[1], Chunbo Duan[1], Shuo Chen[1], Jing Zhang[1], Chunmiao Han [1] & Hui Xu [1] ✉

Despite potential in high-resolution and low-cost displays and lighting, multi-doping structures and low concentrations (<1%) limit repeatability and stability of single-emissive-layer white light-emitting devices. Herein, we report a singly doped white-emitting system of blue thermally activated delayed fluorescence host matrix (CzAcSF) doped by yellow $Cu_4I_4$ cluster ([tBCzDppy]$_2$Cu$_4$I$_4$). CzAcSF:$x$% [tBCzDppy]$_2$Cu$_4$I$_4$ films realize photo- and electro-luminescence colors from cool white to warm white at $x = 20–40$. The external quantum efficiency of 23.5% was achieved at $x = 30$, indicating the record-high efficiency among solution-processed analogs and the largest doping concentration among efficient white light-emitting devices. It shows that di(*tert*-butyl)carbazole moieties in [tBCzDppy]$_2$Cu$_4$I$_4$ provide high-lying excited energy levels at~2.6 eV to mediate energy transfer from CzAcSF (2.9 eV) to coordinated $Cu_4I_4$ (2.2 eV). Our results demonstrate the antenna effect of ligands on optimizing charge and energy transfer in organic-cluster systems and superiority of white cluster light-emitting diodes in practical applications.

White light-emitting diodes are of great importance for developing high-resolution displays and lightings[1,2]. Therefore, diverse luminescent materials are used to realize white electroluminescence (EL), including organic small molecules[3,4], polymers[5], perovskites[6] and clusters[7–9]. Nonetheless, high efficiency, easy processing, low cost and large production are crucial for daily applications of white-emitting devices. Compared to fluorescence (FL)[10], phosphorescence (PH)[11] and thermally activated delayed fluorescence (TADF)[12–15] emitters can utilize 100% electro-generated excitons through either intersystem crossing (ISC) or reverse ISC (RISC). Thus, white organic light-emitting diodes (WOLED) can be superior in energy conservation, but mostly adopted multiple emissive layers (EML) featuring PH and/or TADF dopants dispersed in host matrixes, in order to alleviate intermolecular interaction induced quenching[16].

However, complicated multi-EML structures undoubtedly increase cost, and limit production yield. Therefore, in recent years, single-EML white light-emitting devices have been developed rapidly, whose EMLs commonly compose of host matrixes multiply doped with either three-primary-color or complementary-color dopants[17]. Up to now, single-EML WOLEDs can already realize the top-rank performance, e.g. external quantum efficiency (EQE, $\eta_{EQE}$) beyond 20%[18]. Nevertheless, since the lower bandgaps of long-wavelength dopants, namely yellow or red emitters, their doping concentrations were commonly <1%[19], which can avert excessive charge and energy transfer from blue dopants for achieving the desired white emission[20], which markedly decreased fabrication repeatability and performance stability.

Actually, due to the triplet concentration quenching, doping structure is still indispensable for PH and/or TADF based white light-emitting devices[21]. In this case, the simplest EML composition should be either host matrixes doped with single-molecular white emitters[22], or blue-emitting hosts doped with yellow/orange emitters, namely singly-doping systems[23]. Most of reported single-molecular white emissions were composed of mono-molecule-based blue and excimer-

[1]MOE Key Laboratory of Functional Inorganic Material Chemistry, School of Chemistry and Material Science, Heilongjiang University, 74 Xuefu Road, Harbin 150080, China. ✉e-mail: hxu@hlju.edu.cn

based long-wavelength bands[24,25], but highly sensitive to doping concentrations and host-dopant interactions[26]. In contrast, binary complementary-white systems are significantly more flexible and diverse: (i) both blue and yellow/orange emitters can harvest all electro-generated excitons, including pure PH or TADF and hybrid "PH + TADF" systems[27,28]; (ii) hybrid "PH/TADF + FL" systems, in which PH and TADF components are predominant in triplet utilization[29,30].

However, EL performances of singly-doped systems were markedly lower than those of multi-doping systems, because it is difficult to optimize charge and energy transfer between two emitters for simultaneously achieving good device efficiencies and white emissions. Actually, most of singly-doped WOLEDs also suffered from low doping concentrations (<1%) of yellow emissive dopants[31]. In our previous works, we developed a series of efficient blue TADF emitters with nearly unitary RISC efficiencies, suitable frontier molecular orbital (FMO) energy levels and large steric hindrance, which can enhance exciton utilization efficiencies (EUE, $\eta_{EUE}$)[19,32] and limit excessive carrier and energy transfer to yellow TADF and FL dopants, giving rise to top-rank $\eta_{EQE}$ beyond 20% and doping concentration reaching 2%[18,33]. Nonetheless, further increasing concentration still made the EL emission shift to yellow, accompanied by sharp efficiency decrease. Therefore, it is still a big challenge of developing efficient singly and heavily doped white-emitting systems.

Obviously, the competition and complementarity in exciton allocation of binary white systems with comparable concentrations should be balanced to combine high efficiencies and emission color purity. With this consideration, two key prerequisites should be met: (i) both blue and yellow emitters should have high $\eta_{EUE}$; (ii) carrier and energy transfer should be accurately modulated to achieve efficient white emission at high doping concentrations. In this case, one feasible strategy is incorporating yellow emissive dopant with excited-state characteristics similar but different to those of blue-emitting matrixes. Recently emerging luminescent copper clusters exhibit the unique multi-component excited states, including metal-to-ligand ($^n$MLCT, $n = 1$ and 3 for singlet and triplet, respectively), halide-to-ligand ($^n$XLCT) and intraligand charge transfer ($^n$ILCT) states, locally excited states of ligands ($^n$LLE), and triplet cluster-centered states ($^3$CC)[8,34–38]. Ligand-centered (LC) components, especially intra-ligand charge transfer states, are similar to those of phosphors and TADF molecules, but the CC state is completely different. The charge and energy transfer from blue-emitting host matrixes can be facilitated by the former, but limited by the latter, leading to exciton allocation balance. In this sense, through optimizing LC and CC characteristics and ratios in excited states, luminescent clusters would be competent as yellow/orange emitters in singly and heavily doped white light-emitting systems.

In recent years, on the basis of ligand engineering, we developed a series of bluish green EL $Cu_4I_4$ cubic clusters with the state-of-the-art $\eta_{EQE}$ beyond 20%[39–42]. Nonetheless, despite a single-molecular white-emitting $Cu_4I_4$ cluster also demonstrated[43], there are no white cluster light-emitting devices (CLED) reported until present, since the maximum $\eta_{EQE}$ of yellow emissive $Cu_4I_4$ cubes was still less than 10%[44]. The previous works showed that donor modification can enhance LC components, reduce bandgaps and increase molecular polarity. On the other hand, strengthening metal-metal and metal-counterion interactions in $Cu_4I_4$ cores can enhance the metal-metal-ligand ($^n$MMLCT) and metal-counterion-ligand (($^n$MXLCT)) charge transfer transitions with lower energies, leading to yellow/orange emissions. Meanwhile, ligands would provide suitable energy levels of frontier molecular orbitals (FMO) and LC excited states to tune carrier and energy transfer from blue-emitting matrixes. Obviously, the key challenge for "ligand-mediated" strategy is how to realize stronger cluster-centered interactions and appropriate ligand functionalization for rationally optimizing excited-state characteristics of the clusters.

As a proof of concept, in this contribution, we demonstrate the singly doped white CLEDs with $\eta_{EQE}$ beyond 23% at a high doping concentration reaching 30%. A 2-(diphenylphosphanyl)pyridine (Dppy) with N-P distance of ~2.7 Å is chosen to form a parent $[DPPy]_2Cu_4I_4$ containing a planar $Cu_4$ metallic core with strong inter-copper interactions[45], which significantly enhances metal-metal-ligand (MMLCT) and metal-iodide-ligand (MILCT) charge transfer transitions, but prevents energy transfer from blue-emitting TADF host 0-(4-((4-(9H-carbazol-9-yl)phenyl)sulfonyl)-phenyl)−9,9-dimethyl-9,10-dihydroacridine (CzAcSF). More importantly, through functionalizing Dppy with di-(tert-butyl)-carbazole (tBCz) groups at meta-position, the resulted $[tBCzDPPy]_2Cu_4I_4$ preserves the similar $Cu_4I_4$ configuration (Fig. 1a), but reveals the markedly improved carrier and energy transfer from CzAcSF in its heavily doped spin-coated films, leading to the state-of-the-art photoluminescence (PL) quantum yield (PLQY, $\phi_{PL}$) and $\eta_{EQE}$ beyond 80% and 20%. Theoretical simulation and photophysical and exciton kinetic investigations indicate that tBCz groups contribute deeper occupied molecular orbitals and high-lying excited states comparable to CzAcSF, which make $[tBCzDPPy]_2Cu_4I_4$ competitive with the host at high doping concentrations with respect to the exciton allocation. Therefore, the incorporation of tBCz groups reduces the gap between optoelectronic properties of blue-emitting TADF host and yellow emissive cluster dopant, which is embodied as the antenna effect of tBCzDppy ligands providing the intermediate energy levels to modulate charge and energy transfer processes. These results demonstrate the superiority of cluster emitters in the diversity and tunability of excited-state characteristics, and the importance of ligand engineering for the controllable optimization.

## Results

### Molecular design and structures

With the purpose to enhance Cu-Cu and Cu-I interactions, Dppy unit as planar and rigid pyridine ortho-substituted with flexible diphenylphosphine (DPP) was chosen to form compact bidentate mode, which can form stable coordination with transition metals[46]. Compared to our previous reported biphosphine ligands with P-P distance of 5.8 Å[39,40,42], N-P distance of Dppy unit is only a half, giving rise to reduced inter-copper distances and increased iodine density. On the other hand, introducing an electron-donating tBCz group at meta-position of electron-withdrawing pyridine renders a donor-acceptor (D-A) structure for tBCzDppy. Thus, LC charge transfer (CT) states of $[tBCzDPPy]_2Cu_4I_4$ would be similar to CT states of D-A featured TADF molecules (Fig. 1a). Besides the hole-transporting ability, the first triplet ($T_1$) energy levels of tBCz derivatives (2.5–2.7 eV) are between those of $[Dppy]_2Cu_4I_4$ (2.3 eV) and CzAcSF (2.9 eV). At ortho-position of DPP, the steric hindrance of tBCz group can effectively reduce inter-cluster interaction induced quenching. In this sense, tBCzDppy would endow its cluster with the balanced optoelectronic properties for accurately modulating charge and energy transfer from blue TADF host matrixes in binary white-emitting systems.

tBCzDppy can be readily synthesized through two continuous nucleophilic substitutions from 3-fluoro-2-bromopyridine (Supplementary Fig. 1). The clusters can be simply prepared through refluxing the mixtures of CuI and the corresponding ligands in dichloromethane. Their chemical structures were fully characterized with NMR, MS and elemental analyses. Their accurate configurations were further confirmed by single-crystal X-ray diffraction.

Single crystal structure of $[Dppy]_2Cu_4I_4$ shows that its four Cu atoms are in a rectangle plane with angles of 90° ± 1 and Cu-Cu bond lengths of 2.54 and 2.71 Å, respectively (Fig. 1b). Two coplanar iodine atoms are bonded with two adjacent copper atoms with lengths of 2.59 and 2.67 Å, respectively. The other two iodine atoms are respectively located above and below $Cu_4$ plane, and bonded with all four Cu atoms with lengths of 2.71-2.98 Å. Obviously, in comparison to $Cu_4I_4$ cubes, Cu-Cu and Cu-I interactions in $Cu_4I_4$ core of $[Dppy]_2Cu_4I_4$ are largely enhanced. Furthermore, lengths of Cu-N and Cu-P bonds are 2.01 and 2.22 Å, respectively, which are shorter than Cu-P bond lengths of

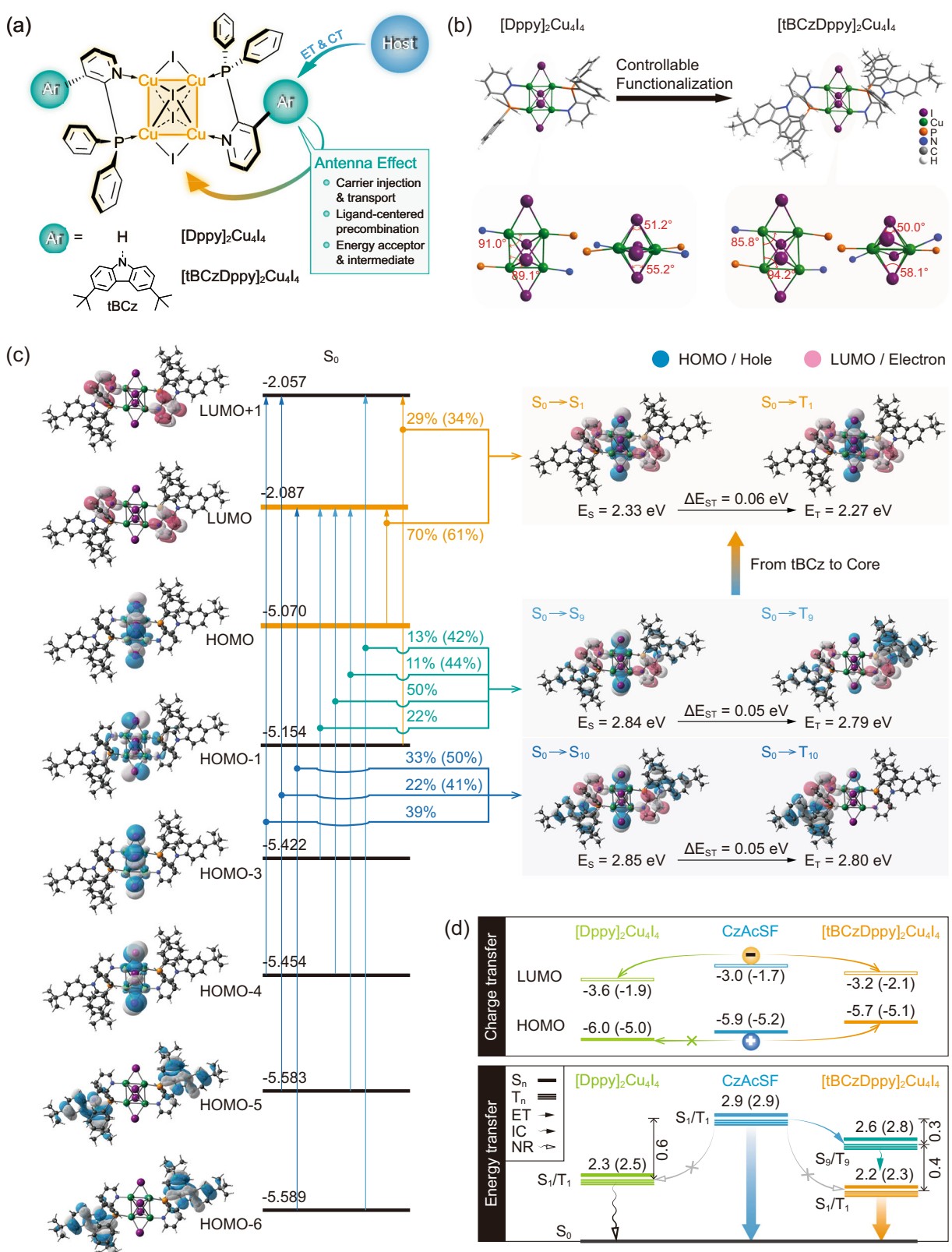

~2.25 Å for phosphine-coordinated $Cu_4I_4$ cubes, reflecting the stronger Cu/I-ligand interactions in $[Dppy]_2Cu_4I_4$. The steric hindrance of tBCz groups at *ortho*-position of DPP groups induces distorted configuration of $[tBCzDppy]_2Cu_4I_4$, whose $Cu_4$ plane becomes a parallelogram with angles of 85.8 and 94.2° and slightly increased bond lengths of 2.57 and 2.73 Å, respectively. In turn, Cu-I bond lengths are respectively decreased to 2.57 Å of coplanar iodine atoms and 2.61 Å of vertex

iodine atoms for the minima. However, its Cu-N and Cu-P bond lengths are unchanged. It indicates that tBCz modification hardly influences Cu/I-ligand interactions, but slightly weakens Cu-Cu interactions and strengthens Cu-I interactions. Consequently, $[tBCzDppy]_2Cu_4I_4$ would obtain the optimized excited-state characteristics featuring enhanced ILCT and MILCT. The single-crystal packing diagram of $[Dppy]_2Cu_4I_4$ reveals inter-cluster edge-to-centroid π-π interactions, which may not

**Fig. 1 | Structures and Electronic properties of the $Cu_4I_4$ clusters. a** Chemical structures of $[Dppy]_2Cu_4I_4$ and $[tBCzDppy]_2Cu_4I_4$ and the design of ligand functionalization with antenna effect in carrier and energy transfer. CT and ET refer to charge and energy transfer, respectively. **b** Single-crystal X-ray diffraction results of $[Dppy]_2Cu_4I_4$ and $[tBCzDppy]_2Cu_4I_4$. The bond angles of $Cu_4I_4$ in coordination skeletons are highlighted from top and front views. **c** Contours and energy levels of ground ($S_0$, left) and selected excited states (right) for $[tBCzDppy]_2Cu_4I_4$ simulated with density functional theory (DFT) and time-dependent DFT (TDDFT) methods. The molecular orbital origin and key parameters of the $S_0 \rightarrow S_1$ and $S_0 \rightarrow T_1$ excitations and di(*tert*-butyl)carbazole (tBCz) involved $S_0 \rightarrow S_9/S_{10}$ and $S_0 \rightarrow T_9/T_{10}$ transitions. "S" and "T" refer to singlet and triplet states. HOMO and LUMO refer to the highest occupied and lowest unoccupied molecular orbitals, respectively. Contours of "holes" and "electrons" and transition parameters of excited states are simulated with natural transition orbital (NTO) analysis. $E$ and $\Delta E_{ST}$ refer to excited-state energy level and singlet-triplet splitting energy. The subscripts of "S" and "T" refer to singlet and triplet states. **d** Proposed mechanisms of facilitated charge (above) and energy transfer (below) between CzAcSF and $[tBCzDppy]_2Cu_4I_4$ mediated by tBCz-contributed energy levels, in comparison to $[Dppy]_2Cu_4I_4$. Data out and in parenthesis are experimental and simulated values, respectively.

only induce collisional quenching, but also reduce solubility (Supplementary Fig. 2). In contrast, no intermolecular interactions can be recognized in packing diagram of $[tBCzDppy]_2Cu_4I_4$ single crystal (Supplementary Fig. 3). Nonetheless, the distance between tBCz and $Cu_4I_4$ cube is only 7.5 Å, which is beneficial to intra-cluster energy and charge transfer from the peripheral ligands to the coordination core (Supplementary Fig. 4). Therefore, the steric hindrance of tBCz groups benefits quenching suppression and energy and charge transfer, and also improves the solubility of $[tBCzDPPy]_2Cu_4I_4$ in common solvents, e.g. chlorobenzene, making device fabrication by solution processing feasible.

## Theoretical simulation

According to density function theory (DFT) simulation on the ground states ($S_0$), the clusters reveal the nearly identical locations of the first five occupied and unoccupied molecular orbitals of the clusters, which are mainly centralized on $Cu_4I_4$ cores and pyridines, respectively (Supplementary Fig. 5). Nevertheless, different to $[DPPy]_2Cu_4I_4$, the sixth (HOMO-5) and seventh (HOMO-6) highest occupied molecular orbitals (HOMO) of $[tBCzDPPy]_2Cu_4I_4$ are thoroughly contributed by its two tBCz groups (Fig. 1c). Furthermore, the energy gap between HOMO-5 and HOMO-4 is only 0.13 eV, which can support fast and effective hole transfer from peripheral tBCz groups to N^P coordinated metallic core, and finally to the HOMO.

Excited-state characteristics of $[Dppy]_2Cu_4I_4$ and $[tBCzDppy]_2Cu_4I_4$ are further investigated with natural transition orbital (NTO) analysis on $S_0 \rightarrow S_n$ and $S_0 \rightarrow T_n$ ($n = 1$–10) excitations (Fig. 1c and Supplementary Figs. 6–8). "Holes" and "electrons" of all singlet and triplet excitations for $[Dppy]_2Cu_4I_4$ and the first eight singlet and triplet excitations for $[tBCzDppy]_2Cu_4I_4$ are respectively dispersed on N^P coordinated $Cu_4I_4$ segments and pyridines. The $S_0 \rightarrow S_1$ and $S_0 \rightarrow T_1$ excitations are attributed to HOMO $\rightarrow$ LUMO transitions, mainly corresponding to mixed M/ILCT. Therefore, the first singlet ($S_1$) and triplet ($T_1$) excited-state energy levels of the clusters are <2.5 eV. Compared to $[Dppy]_2Cu_4I_4$, the electron-donating effect of tBCz groups decreases the $S_1$ and $T_1$ energy levels of $[tBCzDppy]_2Cu_4I_4$ by 0.1 eV. It is noted that the tBCz-localized HOMO-5 and HOMO-6 make significant contributions to "holes" of the $S_0 \rightarrow S_9/T_9$ and $S_0 \rightarrow S_{10}/T_{10}$ excitations. Furthermore, the singlet-triplet splitting energies of these excited states are nearly negligible, which are beneficial to intersystem crossing. Nonetheless, in contrast to the $S_1$ and $T_1$ states with nearly identical distributions, the $S_9/S_{10}$ and $T_9/T_{10}$ states of $[tBCzDppy]_2Cu_4I_4$ are dominantly dispersed on $Cu_4I_4$ core and tBCz groups, respectively, giving rise to more effective spin-orbital coupling[47]. It means these tBCz-involved high-lying excited states likely play important roles in exciton conversion.

It is noteworthy that the experimental energy levels of the FMOs and the first excited states for $[tBCzDppy]_2Cu_4I_4$ reflect the predominant contributions of its tBCz groups (Fig. 1d). In accord to the same locations of their lowest unoccupied molecular orbitals (LUMO), the LUMO energy levels of the clusters are comparable, and ~0.2 eV deeper than that of CzAcSF (Supplementary Fig. 9 and Table S1). On the contrary, different to the HOMO energy level of $[Dppy]_2Cu_4I_4$ at

−6.0 eV, electron-donating effect of tBCz groups induces the HOMO energy level of $[tBCzDppy]_2Cu_4I_4$ also 0.2 eV shallower than that of CzAcSF. Therefore, in opposite to $[Dppy]_2Cu_4I_4$, $[tBCzDppy]_2Cu_4I_4$ can directly form excitons through simultaneous hole and electron capture. On the other hand, the $S_1$ and $T_1$ energy levels of the clusters are about 2.2 eV. In comparison to those of CzAcSF (2.9 eV), the large energy gaps reaching ~0.7 eV prevents host-to-cluster energy transfer. However, tBCz-centralized high-lying $S_9/S_{10}$ and $T_9/T_{10}$ states are ILCT predominant, which are similar to intramolecular CT excited states of CzAcSF. Moreover, the $S_9/S_{10}$ and $T_9/T_{10}$ energy levels of ~2.6 eV are just located in the middle of the $S_1$ and $T_1$ energy levels of CzAcSF and $[tBCzDPPy]_2Cu_4I_4$, therefore they can serve as the intermediate energy levels to facilitate exciton allocation to the cluster through a ladder-like process: energy transfer from $S_1/T_1$ of CzAcSF to $S_9/T_9$ (or $S_{10}/T_{10}$) of $[tBCzDPPy]_2Cu_4I_4$, then to $S_1/T_1$ of $[tBCzDPPy]_2Cu_4I_4$ by internal conversion. In this case, energy gap for each step is reduced to ~0.3 eV, which is appropriate for facile and efficient positive energy transfer.

It is rational that introducing electron-rich tBCz groups on electron-deficient pyridines gives rise to ILCT characteristics analogical to TADF host molecules, therefore, giving rise to the antenna effect for optimizing host-to-cluster charge and energy transfer.

## Photophysical properties

Electronic spectra of the clusters in dilute solutions ($10^{-6}$ M in dichloromethane) consist of ligand-attributed $n \rightarrow \pi^*$ and $\pi \rightarrow \pi^*$ transitions with absorption wavelengths less than 375 nm and long tails in the range of 375–500 nm characteristic of mixed charge transfer states (Fig. 2a and Supplementary Table S1). The absorption peak of $[tBCzDppy]_2Cu_4I_4$ at 328 nm originates from carbazole, reflecting the significant antenna effect of tBCz groups in excited-state population. Similar to the situation of solutions, the absorption of neat $[tBCzDppy]_2Cu_4I_4$ film is stronger than that of $[Dppy]_2Cu_4I_4$ film (Supplementary Fig. 10). The predominant contributions of charge transfer states to radiations result in full widths at half maximum reaching ~150 nm and the peak wavelengths at 548 and 590 nm for PL spectra of $[Dppy]_2Cu_4I_4$ and $[tBCzDppy]_2Cu_4I_4$ films, corresponding to yellowish green and orange emissions. In contrast to near-zero $\phi_{PL}$ of neat $[Dppy]_2Cu_4I_4$ film, $\phi_{PL}$ of neat $[tBCzDppy]_2Cu_4I_4$ film increases by more than 40 folds to 37% (Supplementary Table S1). However, doping $[Dppy]_2Cu_4I_4$ in polymethyl methacrylate (PMMA) matrix can greatly improve $\phi_{PL}$ to 10%, which demonstrates the serious quenching by inter-cluster interactions for $[Dppy]_2Cu_4I_4$ and the self-host characteristics of tBCz moieties in $[tBCzDppy]_2Cu_4I_4$ for quenching suppression. Nevertheless, $\phi_{PL}$ of PMMA:10% $[tBCzDppy]_2Cu_4I_4$ dramatically increases to a state-of-the-art value of 80%.

It is noteworthy that the excitation spectrum of $[Dppy]_2Cu_4I_4$ film is thoroughly attributed to charge transfer states with a main peaks at 370 and 400 nm from MMLCT and MXLCT transitions (Fig. 2a). In contrast, besides the similar charge transfer featured peaks, the main excitation of $[tBCzDppy]_2Cu_4I_4$ film is peaked at 300 and 330 nm characteristic of carbazole, indicating the predominance of tBCz moieties in excited-state population and energy transfer process. Furthermore, excitation-emission mapping of $[Dppy]_2Cu_4I_4$ film

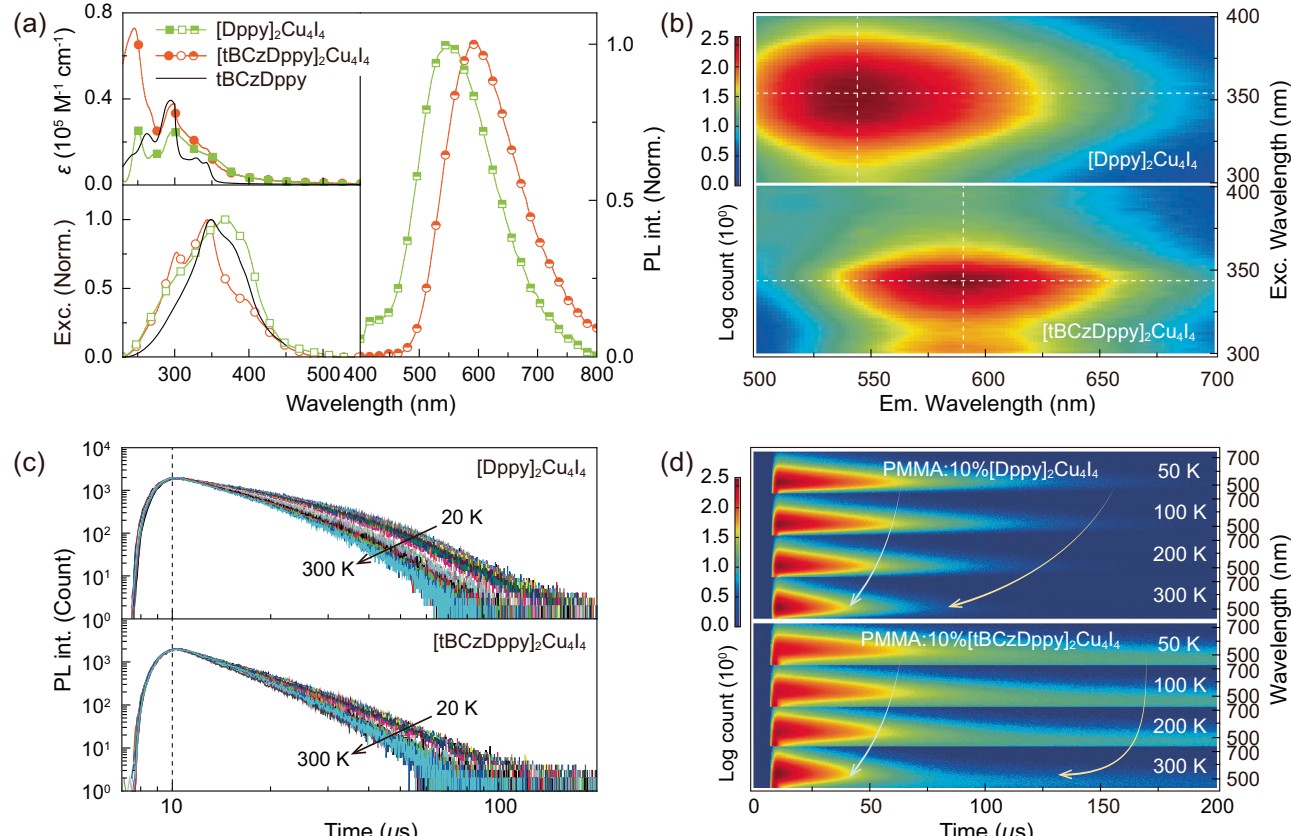

**Fig. 2 | Comparison on photophysical properties of [DPPy]₂Cu₄I₄ and [tBCzDppy]₂Cu₄I₄.** **a** Electronic absorption spectra (upper left) of the clusters in dilute dichloromethane solution ($10^{-6}$ mol L$^{-1}$), in which $\varepsilon$ is molar extinction coefficient, and excitation (lower left) of the neat [DPPy]₂Cu₄I₄, [tBCzDppy]₂Cu₄I₄ and tBCzDppy films, photoluminescence (PL) spectra (right) of the neat [DPPy]₂Cu₄I₄ and [tBCzDppy]₂Cu₄I₄ films. **b** Excitation-Emission mapping of the neat cluster films at room temperature, whose contour centers were highlighted with crosshair points. **c** Temperature dependence of time decays for neat cluster films in the range of 20–300 K with an interval of 10 K. Arrows indicate the variation tendencies. **d** Time-resolved emission spectra (TRES) of neat cluster films at 50, 100, 200 and 300. Light blue and yellow arrows indicate the variation tendencies of concentrated and delayed components, respectively.

reveals wide-range and flat contours, corresponding to mixed MMLCT and MXLCT transitions (Fig. 2b). In contrast, contours of [tBCzDppy]₂Cu₄I₄ film become narrow, and are centralized at 330 nm. Moreover, increasing excitation wavelength makes contours of [tBCzDppy]₂Cu₄I₄ film extended to short wavelengths, reflecting gradually reduced intra-cluster energy transfer from high-lying LC states to MMLCT and MXLCT states. Furthermore, different to temperature-independent excitation spectra of CzAcSF film (Supplementary Fig. 11), excitations of cluster films can be influenced by temperature (Supplementary Fig. 12). Especially, in opposite to reduced MMLCT/MXLCT, tBCz-attributed high-lying LC band of [tBCzDppy]₂Cu₄I₄ film is slightly enhanced at room temperature, indicating the thermally activated character[47,48].

Time decay curves of [Dppy]₂Cu₄I₄ film are doubly exponential, corresponding to two microsecond-scale lifetimes (Fig. 2c). Serious quenching is further demonstrated by markedly decreased lifetimes of [Dppy]₂Cu₄I₄ film at higher temperatures, due to the increased phonon relaxation and inter-cluster interactions (Supplementary Fig. 12c). On the contrary, [tBCzDppy]₂Cu₄I₄ film shows the less temperature-dependent singly exponential time decays, corresponding to lifetimes ~1 µs shorter than those of [Dppy]₂Cu₄I₄ film (Supplementary Table 1), which are consistent with the situation of cluster powders (Supplementary Fig. 13). Therefore, rigid and bulky tBCz moieties largely suppress quenching effects. Furthermore, it can be noticed that there are rising stages before 10 µs in the decay curves, attributed to inter-cluster energy transfer from ILCT and LLE to MMLCT and MXLCT states. Differently, intensity increases of [tBCzDppy]₂Cu₄I₄ film are

more progressive than those of [Dppy]₂Cu₄I₄ film, reflecting the incorporation of tBCz-involved intermediate states.

Time-resolved emission spectra (TRES) of neat and PMMA:10% cluster films reveal the unchanged transition processes during radiation (Fig. 2d and Supplementary Fig. 14). PMMA:10% [Dppy]₂Cu₄I₄ film reveals the slightly elongated decays, owing to the reduced inter-cluster interactions. The almost linearly reverse dependence of decays on temperature verifies the phonon relaxation-induced quenching of [Dppy]₂Cu₄I₄ even in PMMA. In contrast, time decays of PMMA:10% [tBCzDppy]₂Cu₄I₄ are markedly elongated at 50-200 K, but comparable to neat film at 300 K (Supplementary Fig. 14). It displays the radiation from LC states with high rate constants instead of CC states with low rate constants. These results further demonstrate that in addition to suppressed inter-cluster interactions in PMMA, the influence of phonon relaxation on [tBCzDppy]₂Cu₄I₄ is negligible, leading to the state-of-the-art $\phi_{PL}$ of PMMA:10% [tBCzDppy]₂Cu₄I₄.

The feasibility of singly doped white-emitting systems is demonstrated with CzAcSF:$x$% cluster films. Despite the lower $T_1$ energy levels of the clusters (Supplementary Fig. 15), as expected, at high doping concentrations of $x$% = 10–40%, the main peaks in PL spectra of [Dppy]₂Cu₄I₄ based films keep unchanged, which are identical to that of CzAcSF at ~480 nm (Fig. 3a). Obviously, besides the minor spectral extension in the range of 550–750 nm, the large difference between intramolecular charge transfer states of CzAcSF and hybrid MMLCT and MILCT states of [Dppy]₂Cu₄I₄ greatly hinders host-cluster energy transfer. In comparison, tBCzDppy exhibits TADF characteristics similar to CzAcSF, indicating their compatible intermolecular charge

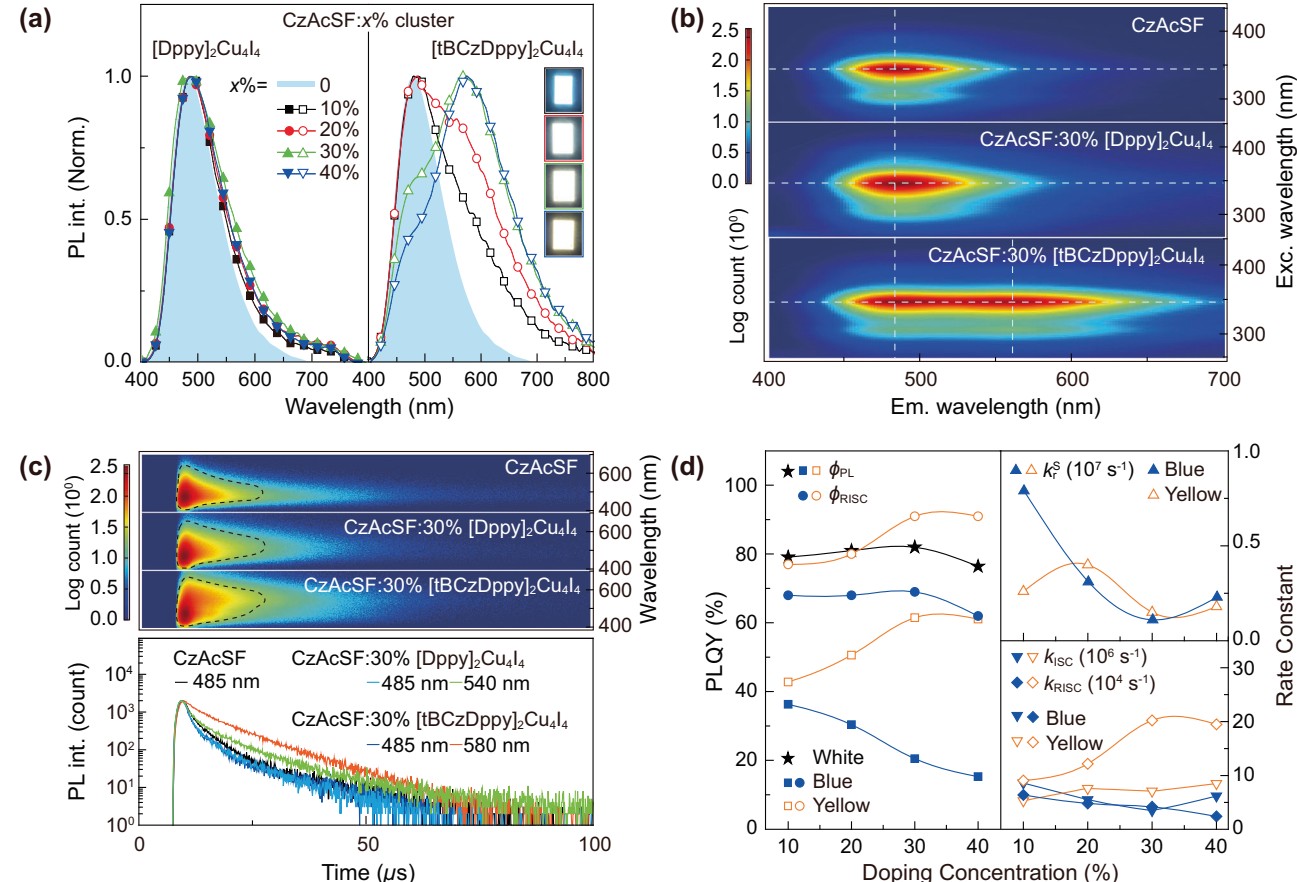

**Fig. 3 | Photophysical properties of CzAcSF:*x*% cluster films. a** PL spectra of CzAcSF:*x*% [DPPy]$_2$Cu$_4$I$_4$ and [tBCzDppy]$_2$Cu$_4$I$_4$ films (*x* = 10, 20, 30 and 40), in which CzAcSF serves as both host and blue thermally activated delayed fluorescence (TADF) emitter. PL spectrum of neat CzAcSF film is also included for comparison, corresponding to *x* = 0. **b** Comparison on excitation-Emission mappings for CzAcSF:*x*% cluster films (*x* = 0 and 30), whose contour centers were highlighted with crosshair points; **c** Comparison on TRES of CzAcSF:*x*% cluster films (*x* = 0 and 30) at room temperature and corresponding time decays at peak wavelengths.

**d** Doping concentration dependence of key transition parameters for CzAcSF:*x*% [tBCzDppy]$_2$Cu$_4$I$_4$ films. $\phi$ refers to quantum yield; while, the subscripts of "PL" and "RISC" refer to photoluminescence and reverse intersystem crossing, respectively; total white emission, blue and yellow components, and blue prompt (PF) and delayed fluorescence (DF) components. *k* refers to rate constant; while, the superscripts of "S" and "T" refer to singlet and triplet states, and the subscripts of "r", "nr" and "ISC" refer to radiation, nonradiation and intersystem crossing, respectively. The blue and yellow components were divided by double-peak fitting.

transfer states (Supplementary Fig. 16). CzAcSF:*x*% [tBCzDppy]$_2$Cu$_4$I$_4$ films display dual-peak emissions containing blue and yellow components originated from CzAcSF matrix (at ~480 nm) and [tBCzDppy]$_2$Cu$_4$I$_4$ dopant (at ~560 nm), respectively. The relative yellow intensity is in direct proportion to *x*%, leading to gradually changed emission colors from cool white at *x* = 20 to warm white at *x* = 40. Obviously, compared to [Dppy]$_2$Cu$_4$I$_4$, incorporation of tBCz moieties in [tBCzDppy]$_2$Cu$_4$I$_4$ establishes the effective host-cluster energy transfer channel, and makes the accurate modulation of white emissions feasible (Supplementary Fig. 17).

Excitation spectra of CzAcSF:30% cluster films are similar to those of neat CzAcSF film at 485 nm (Supplementary Figs 18 and 19). But, different to [Dppy]$_2$Cu$_4$I$_4$ doped film, excitation spectra of CzAcSF:30% [tBCzDppy]$_2$Cu$_4$I$_4$ at 570 nm reveal significant high-lying LC component in addition to CzAcSF-attributed bands, manifesting the intermediate effect of tBCz moieties. The excitation-emission mapping of CzAcSF:30% cluster films indicates that the contour of [DPPy]$_2$Cu$_4$I$_4$ doped film is nearly identical to that of neat CzAcSF film; while, despite the different contour of [tBCzDPPy]$_2$Cu$_4$I$_4$ based film, the excitation ranges and profiles of the non-doped and doped films are totally the same (Fig. 3b). It means as the majority in the films, CzAcSF is predominant in excitation; while, emissions from the clusters are highly dependent on energy transfer from CzAcSF, which is consistent with doping concentration dependent time decays of yellow components

from [tBCzDppy]$_2$Cu$_4$I$_4$ doped films (Supplementary Fig. 20). So, tBCz moieties undoubtedly and greatly improve the energy transfer to [tBCzDppy]$_2$Cu$_4$I$_4$.

TRES contour of CzAcSF:30% [Dppy]$_2$Cu$_4$I$_4$ is only broadened in comparison to that of neat CzAcSF film (Fig. 3c and Supplementary Fig. 21). Differently, CzAcSF:30% [tBCzDppy]$_2$Cu$_4$I$_4$ reveals yellow-predominant TRES contour, indicating the predominance of the cluster in exciton harvesting. Time decays show that after cluster doping, the lifetimes of CzAcSF are reduced. Furthermore, $\phi_{PL}$ of CzAcSF:30% [tBCzDppy]$_2$Cu$_4$I$_4$ reaches 82%, which is comparable to those of neat CzAcSF and [tBCzDppy]$_2$Cu$_4$I$_4$ films, and four folds of that of [Dppy]$_2$Cu$_4$I$_4$ doped analog (Supplementary Table 2). In this sense, the blue lifetime decrease is due to energy transfer and quenching for [tBCzDppy]$_2$Cu$_4$I$_4$ and [Dppy]$_2$Cu$_4$I$_4$ based films, respectively. Meanwhile, yellow decay of [tBCzDppy]$_2$Cu$_4$I$_4$ based film is markedly longer than that of [Dppy]$_2$Cu$_4$I$_4$ based analog. It is noteworthy that the relative contributions of [Dppy]$_2$Cu$_4$I$_4$ to concentrated and delayed components are comparable; while, the yellow proportion in delayed component of CzAcSF:30% [tBCzDppy]$_2$Cu$_4$I$_4$ is markedly larger than that in concentrated component, reflecting the predominance of triplet energy transfer. Furthermore, different to consistent excitation and emission centers of CzAcSF:30% [Dppy]$_2$Cu$_4$I$_4$, the facilitated energy transfer from CzAcSF to [tBCzDppy]$_2$Cu$_4$I$_4$ is demonstrated by the center shift from excitation on the host to emission from the

cluster (Supplementary Fig. 22). Actually, NTO results show that compared to the $S_9$ and $S_{10}$ states with significant contributions from $Cu_4I_4$, the $T_9$ and $T_{10}$ states of $[tBCzDppy]_2Cu_4I_4$ are typical $^3$ILCT, whose "holes" and "electrons" are respectively localized on tBCz donors and pyridine acceptors, which are highly similar to intramolecular charge transfer of CzAcSF. Consistently, sliced TRES of CzAcSF:30% $[tBCzDppy]_2Cu_4I_4$ indicates a transition band occurring in the time range of 12–40 μs, corresponding to the intermediate triplet states located on tBCz groups (Supplementary Fig. 23). Thus, tBCzDppy serves as the antenna to facilitate energy transfer in CzAcSF:*x*% $[tBCzDppy]_2Cu_4I_4$, through providing its $T_9$ and $T_{10}$ states as intermediate states.

In contrast to $\phi_{PL}$ of CzAcSF:*x*% $[Dppy]_2Cu_4I_4$ reversely proportional to *x*%, CzAcSF:*x*% $[tBCzDppy]_2Cu_4I_4$ reveal stable $\phi_{PL}$ reaching ~80% in the range of 10–40% (Supplementary Table 2), owing to the complementarity of the host and cluster in radiation (Fig. 3d and Supplementary Fig. 24). It is noteworthy that different to comparable prompt and delayed quantum efficiencies of $[Dppy]_2Cu_4I_4$ doped films, the incorporation of $[tBCzDppy]_2Cu_4I_4$ induces gradually decreased blue proportions and sharply increased yellow proportions in delayed quantum efficiencies, further verifying the predominance of triplet states in energy transfer. It is noted that at *x* = 30, rate constants

of blue prompt and delayed components and ISC and RISC transitions for $[Dppy]_2Cu_4I_4$ and $[tBCzDppy]_2Cu_4I_4$ doped films are comparable, but blue singlet rate constants of the latter are markedly larger than those of the former. Furthermore, yellow delayed rate constant of CzAcSF:30% $[tBCzDppy]_2Cu_4I_4$ are even higher, but its yellow singlet rate constants are the lowest, in accord to the significant contribution of $[tBCzDppy]_2Cu_4I_4$ in triplet utilization. Besides higher yellow RISC efficiencies ($\phi_{RISC}$) beyond 90%, yellow ISC and RISC rate constants of CzAcSF:30% $[tBCzDppy]_2Cu_4I_4$ are also significantly larger than blue components. Therefore, $[tBCzDppy]_2Cu_4I_4$ is crucial for singlet-triplet conversion and triplet harvesting.

## Electroluminescent properties

Compared to $[Dppy]_2Cu_4I_4$, tBCz modification improves the film formability, morphological stability and matrix compatibility of $[tBCzDppy]_2Cu_4I_4$, making device fabrication by solution processing feasible (Supplementary Fig. 25). A simple trilayer structure was adopted to fabricate CLEDs using CzAcSF:*x*% clusters as EMLs through spin coating at heavy doping concentration of *x*% = 10–40% (Fig. 4a). The intrinsic EL properties of the clusters were further characterized by using a conventional bipolar host bis-4-((*N*-carbazolyl)phenyl)-phenylphosphine oxide (BCPO) (Supplementary Fig. 26). Compared to

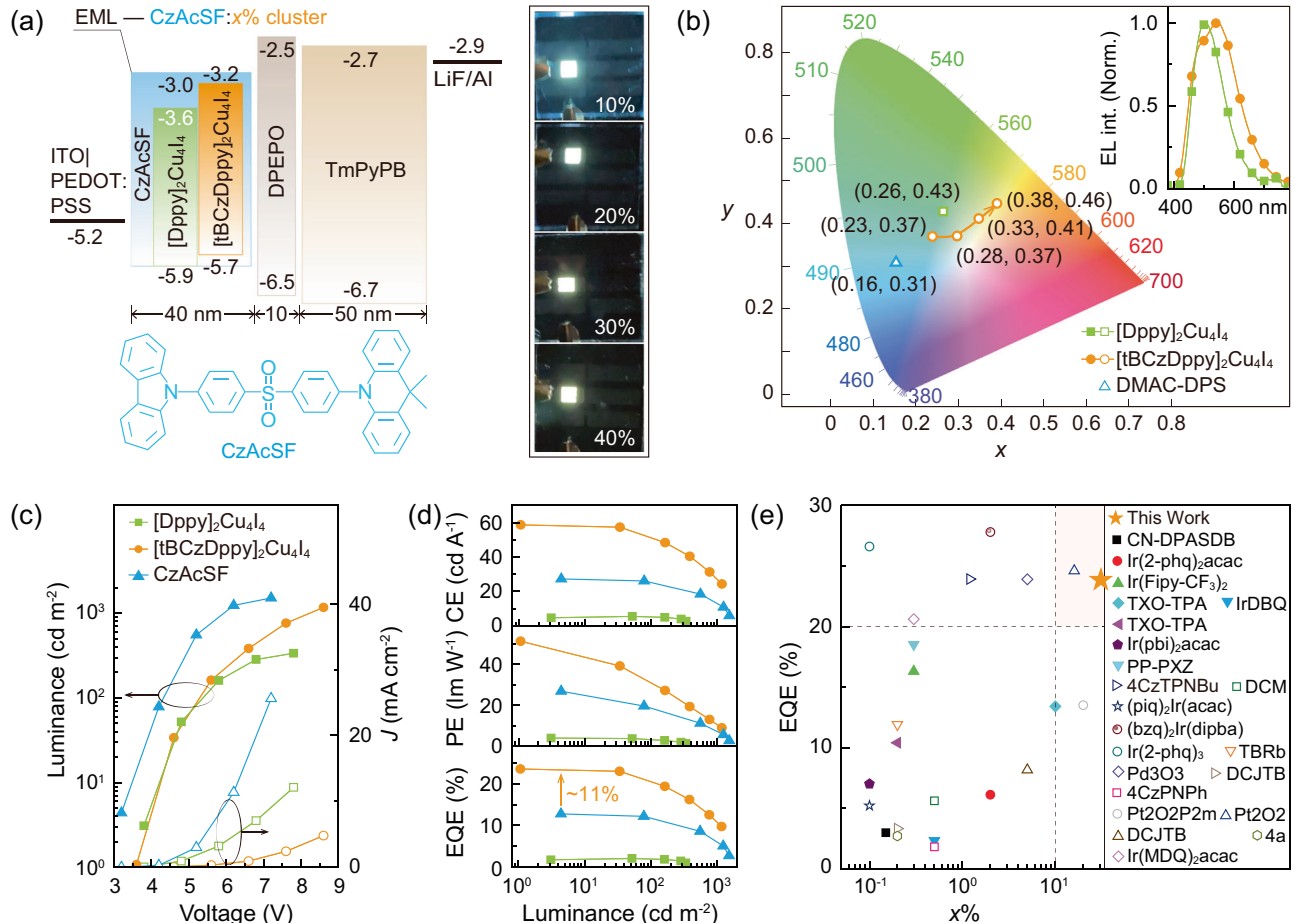

**Fig. 4 | Electroluminescence (EL) performance of spin-coated CzAcSF:*x*% cluster based devices. a** Device configuration of the cluster light-emitting diodes (CLED), and chemical structure of CzAcSF serving as blue TADF host. **b** Commission Internationale de lEclairage (CIE) coordinates of the devices on CIE 1931 chromatic panel and corresponding EL spectra of CLEDs (insets). Different to *x* = 30 for $[DPPy]_2Cu_4I_4$, arrow indicates the variation of CIE coordinates at *x* = 10–40 for $[tBCzDPPy]_2Cu_4I_4$. The CIE coordinates of spin-coated device based on neat CzAcSF are included for comparison. **c** Current density-voltage-luminescence curves and

**d** Efficiencies vs. luminance relationships of CzAcSF:30% cluster based devices. The data of neat CzAcSF based control device are also presented for comparison. **e** Summary of external quantum efficiencies (EQE) and optimal doping concentrations (*x*%) for representative singly doped white light-emitting devices. Solid and hollow symbols indicate the device fabrication through spin coating and *vacuum* evaporation, respectively. For detailed comparison on EL performance, please see Table S5.

sky-blue device using neat CzAcSF as EML (Supplementary Fig. 27a), 30% [Dppy]$_2$Cu$_4$I$_4$ doped device showed greenish emission (insets in Fig. 4b and S28a), corresponding to CIE coordinates of (0.26, 0.43) increased by (0.10, 0.12) (Fig. 4b). In contrast, devices of CzAcSF:$x$% [tBCzDppy]$_2$Cu$_4$I$_4$ revealed dual-peak EL emissions with gradually enhanced yellow components (insets in Supplementary Fig. 29a), inducing colors changed from cool white, over pure white to warm white along with increasing $x$% (inset in Fig. 4a). CIE coordinates of CzAcSF:$x$% [tBCzDppy]$_2$Cu$_4$I$_4$ based devices were basically along Planckian locus (Fig. 4b), verifying their favorable white color purity. At $x = 30$, color rendering index (CRI) can reach to 81, which is high among singly doped white devices. The corresponding correlated color temperatures (CCT) of 7632, 5675 and 4532 K at $x = 20$–40 were in accord to standard illuminants of D75, D50 and CWF, respectively, demonstrating the competence for artificial lightings (Supplementary Table 3). Compared to [Dppy]$_2$Cu$_4$I$_4$ based analogs, [tBCzDppy]$_2$Cu$_4$I$_4$ endowed its devices with the lower driving voltages and higher luminance (Supplementary Tables 3 and 4). Therefore, tBCz modification made [tBCzDppy]$_2$Cu$_4$I$_4$ involved in electrical processes, and improved exciton formation and radiation.

It shows that spin-coated devices of neat CzAcSF can achieve the maximum efficiencies of 27.3 cd A$^{-1}$ for current efficiency (CE, $\eta_{CE}$), 26.8 lm W$^{-1}$ for power efficiency (PE, $\eta_{PE}$) and 12.7% for $\eta_{EQE}$, respectively (Supplementary Fig. 27b and Supplementary Table 3). In addition to markedly decreased luminance (Fig. 4c and Supplementary S28a and Table 4), despite slightly changed spectra, the device efficiencies of CzAcSF:$x$% [Dppy]$_2$Cu$_4$I$_4$ were largely reduced and in reverse proportion to $x$%, reflecting the worsened exciton quenching (Fig. 4d and Supplementary Fig. 28b and Table 4). In contrast, incorporation of [tBCzDppy]$_2$Cu$_4$I$_4$ led to nearly doubled efficiencies with the maxima of 58.7 cd A$^{-1}$, 51.2 lm w$^{-1}$ and 23.5% at $x = 30$, which are the record values of white CLEDs reported so far, and also among the best results for all-color CLEDs (Fig. 4d and Supplementary Table 3). It is noted that the efficiencies firstly increased to reach the maxima at $x = 30$, and then decreased (Supplementary Fig. 29b). The repeatability of CzAcSF:30% [tBCzDppy]$_2$Cu$_4$I$_4$ based devices were verified by their stable efficiencies (Supplementary Fig. 30). Although TADF feature of CzAcSF and simple structures of the spin-coated devices could reduce duration times, the efficiency roll off was only 12.7% at 100 cd m$^{-2}$, which was high among solution-processed devices, reflecting the modified carrier recombination and suppressed exciton quenching (Supplementary Fig. 31).

Compared to nearly non-emissive [Dppy]$_2$Cu$_4$I$_4$ based analogs with luminance ≤30 cd m$^{-2}$ and $\eta_{EQE} \leq 0.3$%, yellow devices of BCPO:$x$% [tBCzDppy]$_2$Cu$_4$I$_4$ realized the state-of-the-art efficiencies of 65.3 cd A$^{-1}$, 38.5 lm w$^{-1}$ and 22.3% at $x = 20/30$, corresponding to -100% internal quantum efficiency and exciton utilization ratio, which are the record-high values among yellow/orange CLEDs reported so far (Supplementary Figs 32 and 33 and Table 4). It shows that the performances of [tBCzDppy]$_2$Cu$_4$I$_4$ in BCPO and CzAcSF hosted devices were almost equal (Supplementary Tables 3 and 4). Furthermore, EL efficiencies of CzAcSF:30% [tBCzDppy]$_2$Cu$_4$I$_4$ were markedly higher than those of neat CzAcSF, therefore the incorporation of [tBCzDppy]$_2$Cu$_4$I$_4$ in exciton utilization actually alleviated the exciton quenching in CzAcSF matrix. In this sense, the exciton utilization ability of the cluster dopants is the key determinant of the performances for singly doped white CLEDs.

Compared to singly-doped white light-emitting devices reported so far, 30% is the highest doping concentration, which can undoubtedly improve device repeatability (Fig. 4e and Supplementary Table 5). Furthermore, different to *vacuum*-evaporated analogs, no spin-coated devices can achieve $\eta_{EQE}$ of 20% previously. tBCz modification simultaneously improves the morphological and electrical properties and exciton utilization for [tBCzDppy]$_2$Cu$_4$I$_4$, therefore facilitating host-cluster energy transfer, enhancing device repeatability and alleviating exciton quenching. It is convincing that the unique excited-state

characteristics of [tBCzDppy]$_2$Cu$_4$I$_4$ is well complementary with TADF feature of CzAcSF, leading to the desired heavily and singly doped white light-emitting devices with the state-of-the-art efficiencies.

Temperature-dependent EL spectra show the nearly unchanged contribution of [Dppy]$_2$Cu$_4$I$_4$ to whole emissions of CzAcSF:30% [Dppy]$_2$Cu$_4$I$_4$ in the range of 50–300 K (Fig. 5a). In contrast, for CzAcSF:30% [tBCzDppy]$_2$Cu$_4$I$_4$ based devices, yellow component from the cluster is slightly in direct proportion to temperature, which is opposite to PL situation. It is because increasing temperature can facilitate the carrier capture by [tBCzDppy]$_2$Cu$_4$I$_4$ in EL process, due to the enhanced molecular motion, which verifies the intermediate effect of tBCz moieties in carrier transfer from CzAcSF to the cluster. Compared to neat CzAcSF based control devices, time-resolved EL emission spectra (TREES) of CzAcSF:30% clusters shows that in addition to the decrease of concentrated component, delayed component of [Dppy]$_2$Cu$_4$I$_4$ doped devices was elongated; while, on the contrary, TREES of [tBCzDppy]$_2$Cu$_4$I$_4$ based devices is more concentrated, accompanied by markedly reduced delayed component (Fig. 5b). Consistently, EL lifetimes of neat CzAcSF based devices were shorter than those of [Dppy]$_2$Cu$_4$I$_4$ doped analogs, but longer than those of [tBCzDppy]$_2$Cu$_4$I$_4$ doped analogs. Consequently, [Dppy]$_2$Cu$_4$I$_4$ doping worsened exciton quenching; while, incorporating [tBCzDppy]$_2$Cu$_4$I$_4$ facilitated exciton radiation, therefore simultaneously suppressed quenching.

Sliced TREES show that nearly identical to the situation of neat CzAcSF based analogs, the contour centers at recombination and decay stages of [Dppy]$_2$Cu$_4$I$_4$ doped devices corresponded to emissions from CzAcSF matrix, reflecting the predominance of CzAcSF in both carrier and exciton capture (Fig. 5c). However, the carrier recombination in CzAcSF:30% [Dppy]$_2$Cu$_4$I$_4$ is about 2 μs earlier than that in neat CzAcSF based devices. Considering the deeper LUMO of [Dppy]$_2$Cu$_4$I$_4$ than CzAcSF, the latter likely facilitated carrier injection in EMLs. On the contrary, for CzAcSF:30% [tBCzDppy]$_2$Cu$_4$I$_4$ based devices, carriers were firstly recombined on CzAcSF matrix, and then transferred to [tBCzDppy]$_2$Cu$_4$I$_4$ after ~3 μs. Similarly, in decay stage, emission was firstly from CzAcSF matrix, and energy transfer to [tBCzDppy]$_2$Cu$_4$I$_4$ rendered yellow emission. The IV characteristics of nominal single-carrier transporting devices show that that the carrier transporting abilities of [tBCzDppy]$_2$Cu$_4$I$_4$ are 1–2 orders of magnitude stronger than those of [Dppy]$_2$Cu$_4$I$_4$ (Supplementary Fig. 34). Despite the predominance of CzAcSF in electrical process, hole-only current densities of CzAcSF:30% [tBCzDppy]$_2$Cu$_4$I$_4$ based devices were still larger than those of CzAcSF:30% [Dppy]$_2$Cu$_4$I$_4$ based analogs. It manifests tBCz groups establish hole injection and transportation channels from CzAcSF matrix to the cluster dopants. Obviously, in accord to PL results, tBCz moieties significantly enhanced carrier and energy transfer from CzAcSF to the cluster.

Therefore, EL processes of CzAcSF:$x$% [Dppy]$_2$Cu$_4$I$_4$ and [tBCzDppy]$_2$Cu$_4$I$_4$ based devices were different (Fig. 5d). For the former, the equal HOMO energy levels of CzAcSF and [Dppy]$_2$Cu$_4$I$_4$ and their singlet and triplet gaps of -0.6 eV induced the predominance of CzAcSF in exciton allocation. [Dppy]$_2$Cu$_4$I$_4$ with low luminescent efficiency in turn worsened exciton quenching. In opposite, although carriers were firstly captured by CzAcSF matrix, the appropriate energy gaps between FMOs and singlet/triplet states of the host and [tBCzDppy]$_2$Cu$_4$I$_4$ greatly facilitated the exciton allocation for the balance of sky-blue and yellow components. In this sense, the antenna effect of tBCz moieties in carrier and energy transfer is crucial for combining high efficiencies and white color purity for singly and heavily doped TADF-cluster hybrid systems.

## Discussion
A N^P hybrid coordinated Cu$_4$I$_4$ cluster [tBCzDppy]$_2$Cu$_4$I$_4$ featuring Cu$_4$ parallelogram with strengthened inter-copper interactions are developed, whose enhanced MMLCT and MXLCT not only result in

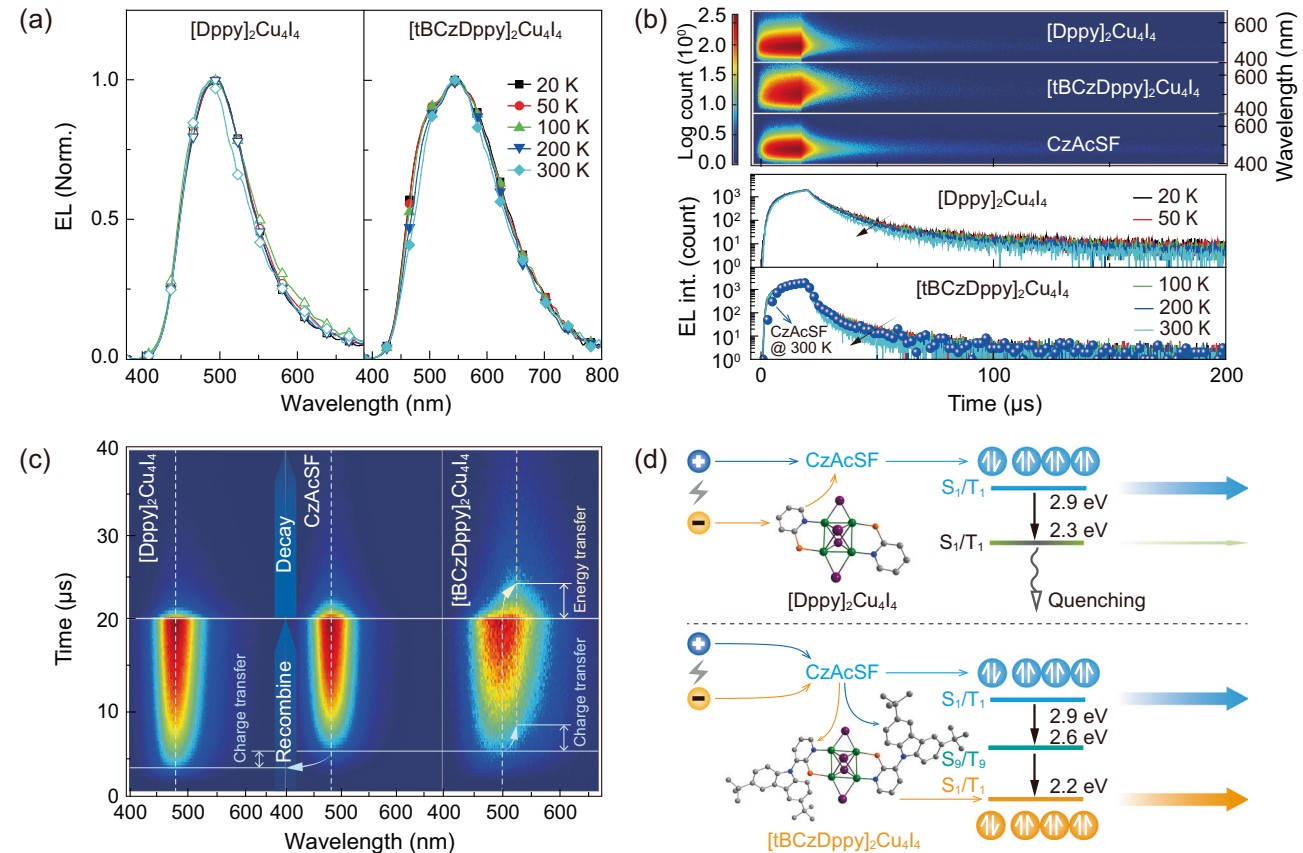

**Fig. 5 | Exciton kinetics of CzAcSF:30% cluster based devices. a** Temperature dependence of EL spectra for the CLEDs at 20, 50, 100, 200 and 300 K. **b** Time-resolved electroluminescence emission spectra (TREES) of the CLEDs at room temperature and temperature dependence of corresponding time decays at peak wavelengths in the range of 20–300 K. TREES and time decay of CzAcSF based control device are included for comparison. **c** Sliced TREES contours of the CLEDs at the stages of recombination (0-20 μs) and decay beginning (20–40 μs), in comparison to the contour of CzAcSF based control device. **d** Comparison on different exciton allocation processes through carrier and energy transfer from CzAcSF for [DPPy]₂Cu₄I₄ and [tBCzDppy]₂Cu₄I₄ based devices.

yellow emissions, but also prevent direct energy transfer from pure-organic TADF host CzAcSF to coordination segment. However, tBCz moieties provide deeper occupied molecular orbitals and high-lying excited states matching with CzAcSF to support effective host-cluster charge and energy transfer, and simultaneously facilitate radiation of the coordination segment through internal conversion. This ligand antenna effect accurately optimizes the exciton allocation between CzAcSF and [tBCzDppy]₂Cu₄I₄, leading to high-purity white PL and EL emissions at a doping concentration of 30% with the state-of-the-art $\phi_{PL}$ and $\phi_{EQE}$ of 82% and 23.5%, respectively. These results not only demonstrate the superiorities of cluster emitters for practical white lighting applications, but also indicate the flexible manipulation of excited-state properties for cluster molecules and exciton processes in organic-cluster hybrid systems through ligand engineering.

## Methods

### Synthesis of copper iodide clusters

1 mmol of ligands and 2 mmol of CuI were dissolved in dichloromethane (5 mL) and stirred for 4 h at room temperature to afford crude clusters as precipitates, which were recrystallized with dichloromethane and ether to afford pure clusters as yellow crystals. Detailed experimental procedures and structural characterization (Supplementary Figs. 35–42) are provided in the Supplementary Information.

### Electroluminescence analysis

The devices were fabricated through spin-coating for emissive layers and vacuum evaporation for electron-transporting layers and cathodes, respectively. A system composed of a Keithley 4200 source meter, a calibrated silicon photodiode and a PR655 spectrum colorimeter was used to measure voltage-current density-luminance characteristics and electroluminescence spectra. Transient and temperature-dependent electroluminescence spectral measurements were performed with FLS 1000 by incorporating a Tektronix AFG3022G function generator.

## Data availability

The authors declare that the data generated in this study are provided in Supplementary Information. The X-ray crystallographic data of [tBCzDppy]₂Cu₄I₄ generated in this study have been deposited in the Cambridge Crystallographic Data Centre (CCDC) under deposition number 2298772 [https://doi.org/10.5517/ccdc.csd.cc2cwf4v].

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

## Acknowledgements

This study was supported by National Natural Science Foundation of China (92061205 (H.X.), 22325502 (H.X.), 62175060 (J.S.), 52273173 (C.H.), 22005088 (J.Z.)), Changjiang Scholar Program of Chinese Ministry of Education (Q2021256 (C.H.)) and Natural Science Foundation of Heilongjiang Province (YQ2020B006 (J.Z.) and YQ2022B010 (C.D.)).

## Author contributions

H.X. conceived the projects. J.S., N.L., Z.G. and C.Z. synthesized the clusters and performed the measurements. C.D. assisted the device fabrication and measurement, Y.M. and S.C. assisted structure characteristics, H.X., J.S., J.Z. and C.H. analysed the data, and J.S. and H.X. wrote the paper. All authors commented on the manuscript.

## Competing interests

The authors declare no competing interests.
