## [Peer Review File · Nature Communications]

Ligand-Mediate Exciton Allocation Enables Efficient Cluster-Based White Light-Emitting Diodes via Single and Heavy DopingREVIEWER COMMENTS

Reviewer #1 (Remarks to the Author):

In this paper, Xu et al. designed two copper-iodide clusters of yellow emission, named [Dppy]₂Cu₄I₄ and [tBCzDppy]₂Cu₄I₄. The latter one was doped into a blue-emissive TADF host, named CzAcSF, and fabricated a white-emission CLED device with EQE up to 23.5% at a high doping concentration of 30%. To explain the outstanding performance, photophysical measurement, and quantum chemical calculation were carried out by the authors. The results showed that the tBu-carbazole-modified ligand (tBCzDppy) provides deeper occupied orbitals and high-lying excited states, which enhance the efficiency of charge transfer and energy transfer. These factors lead to the high EQE of the device. The structure-relationship is clear and the molecular design strategy is instructive, so I recommend this paper to be published on Nat. Commun. after addressing the following comments:

1. Page 1, Line 15, “yellow Cu₄I₄” or “yellow emissive Cu₄I₄”?
2. Page 4, Line 94, “high doing concentration” should be “high doping concentration”.
3. Page 12, Line 369 “Copper iodide cluster synthesis”. Please provide the detailed synthesis process utilized to obtain the clusters. Additionally, include characterizations of the Cu₄I₄ clusters, including NMR spectroscopy, mass spectrometry (MS), and elemental analysis either in the main text or in the SI.
4. Supporting information, page 21, line 168 “table S1”, the energy of S1 was obtained by the edge of absorption spectra, but the energy of T1 was obtained by the emission spectra. I’m not sure how to confirm the 0-0 transition energy of the T1-S0 state. Please check and revise accordingly.
5. Supporting Information, page 22, line 177 “table S2”, please provide equations that you used to calculate those rate constants of each photophysics process.
6. In the main text, please provide further descriptions of the reasons why the authors choose to modify the group on the ortho position of the carbon in the Dppy ligand, as well as the reason for selecting tert-butyl carbazole as the modification.
7. Supporting Information, page 21, line 171 “table S1”, the words “[h] in PMMA matrix” but it is not referred to in the table. Please check it carefully.
8. Supporting Information, page 22, line 177 “table S2”, please tell the details of how you measure the PLQY of [b] blue and [c] yellow components of CzAcSF:x% cluster films, respectively.

Reviewer #2 (Remarks to the Author):

This is an interesting study that presents singly doped white LEDs with EQE beyond 23% at extremely high doping concentrations. Exciton kinetics have been investigated and a relation for the higher efficiency has been fashioned reasonably. However, the discussion is insufficient and not clear enough. In spite of excellent EQE of the white LED reported with exciton studies, the materials reported here and the current state of work is inadequate to be published in this Nature sister journal.

Here are some other concerns:

Researchers did not conduct any studies to further understand the morphological properties of the synthesized materials. Since the devices were fabricated using solution processing technique, morphology is among the crucial factors affecting device performance and stability. The researchers did not investigate the charge transport properties of the cluster materials and doped TADF films. No device lifetime (T50 and T90), stability (in different conditions) and reproducibility of the fabricated devices were discussed in the manuscript.

The solvent(s) used for device fabrication were not stated in the manuscript.

There is no discussion on the turn-on voltage (V_{on}) differences observed in different devices. In addition, in the table S3 and S4, the voltages measured for 1 cd/m² were displayed. However, in figures S17-22, no intercept was observed at 1 cd/m² which is very bizarre. In Figure 4c and S18a, voltages used for devices measurements were random and are not in order. This is the scenario in other device JVL graphs but it is too prominent in these two graphs except Figure S20. Figure S20 itself is very erratic and not following any trend. Even so, no explanation for such uncommon performance observed was discussed.

Overall, despite an outstanding performance observed, the explanation for such performance was not supported properly. Just using DFT results from Gaussian09 and exciton kinetics, the authors tried explaining the improved performance of the devices. A lot of effort has been put to reason the high EQE but even that falls short and rest was either ignored or skipped. The device performance is unorganized and not reliable.

Reviewer #3 (Remarks to the Author):

In this work, the authors selected yellow-emitting copper clusters and a blue TADF host CzAcSF to form the white-emitting devices. Although there were reports of white OLEDs and PeLEDs, I agree with the authors that clusters based on eco-friendly cheap metals can provide the important alternatives for the large-scale applications, since the high rigidity of clusters can limit the J-T distortion of mononuclear complexes. I noticed that Yao et al. recently reported a warm white-emitting device of Cu₂I₂ clusters with broad emissions, which was obviously different to this work regarding to material design and photophysical investigation (Nat Photon 18, 200-206 (2024)). The performance reported in this work was also improved. What I am interested in is the heavy doping for realizing white emission reported in this work, which was thoroughly different to all previously reported white-emitting systems. This finding may discover one advantage of cluster emitters. The authors organized the experiments and investigations carefully to figure out the fundamentals behind the unique photophysical processes in CzAcSF:Cu₄I₄ cluster films and the key factors of carrier and energy transfer. The EQE of 23.5% achieved at 30% doping concentration by the solution-processed devices indicated the potential of white cluster light-emitting devices. Therefore, I would like to recommend the publication of this work in Nature Communications.

Some minor revisions would be necessary:

1. It was a good idea to use copper clusters as long-wavelength emitters for binary white-emitting systems, considering the sustainability and low cost of copper, since previous works were based on noble-metal complexes containing iridium and platinum, etc. For example, Forrest et al. reported the white OLEDs based on Ir(ppy)₃ and PQIr (Nature 440, 908-912 (2006)). But, I still wonder are there any other fundamental advantages of the copper clusters for white-emitting applications.
2. On page 7, paragraph 3, the authors claimed that the emission duration of [Dppy]₂Cu₄I₄ film decreased more significantly at higher temperatures, in comparison to [tBCzDppy]₂Cu₄I₄ film, because the rigid and bulky tBCz groups inhibited the quenching effect. Since tBCz groups also modified the solubility and film formability, the film morphology of [tBCzDppy]₂Cu₄I₄ might be different with that of [Dppy]₂Cu₄I₄, the authors are suggested to add and discuss the time decay curves of the clusters in powder.
3. On page 8, paragraph 3, CzAcSF:x% [tBCzDppy]₂Cu₄I₄ film exhibited dual-peak emission, and the relative intensity of yellow was proportional to x%, resulting in emission color change from cool white to warm white. Since the authors claimed the importance and limitation of host-cluster energy transfer, the dependence of emission color on the smaller x% change would reflect the improved energy transfer from CzAcSF to [tBCzDppy]₂Cu₄I₄.
4. The authors claimed that different to Dppy, tBCzDppy may have the TADF characteristics due to its donor-acceptor structure. I suggest them to compare the photophysical properties of Dppy and tBCzDppy to support it, especially temperature-related decays.
5. On page 10, line 305, "At x =30, color rendering index (CRI) can reach to 81" was different to the CRI value of 72 in Table S3. The authors should carefully check the data.

Reviewer #4 (Remarks to the Author):

This work reported a novel single-doped white light emission system based on blue TADF host (CzAcSF) and yellow Cu₄I₄ cluster dopant ([tBCzDppy]₂Cu₄I₄). The EL emission changed from cold white to warm white through tuning doping concentration of 20-40%, owing to the limited host-cluster energy transfer. When the doping concentration was 30%, the EQE of the solution-processed devices reached 23.5%. The advantage of singly cluster doped white-emitting device in heavy doping is important for applications. The authors made a systematic investigation and rational explanation on the improved energy transfer between cluster and CzAcSF, through high-lying excited states from carbazole moieties. The authors demonstrated the main conclusions with solid evidences, which would be important to understand the unique excited-state properties of cluster emitters. This work can be published in Nature Communications after suitable revisions.

1. I have noticed that in recent years, there have been some works used clusters as emissive layers to prepared electroluminescent devices, most of which showed blue or yellow emissions, and the reports of white devices based on clusters were only few (Nat. Commun. 2023, 14, 2901; Angew. Chem. Int. Ed. 2021, 61, e202213826). Please briefly introduce the main challenge of developing cluster based white light-emitting devices.

2. Dppy is a common ligand for cluster preparation. Wang et al prepared gold clusters based on Dppy (J. Am. Chem. Soc. 2013, 135, 16184). Therefore, are there any advantages of this kind of ligands in comparison to triphenylphosphine, regarding to EL cluster design.

3. In Fig 2 (d), the time-resolved emission spectra (TRES) of PMMA : 10% cluster films revealed that the effect of phonon relaxation on [tBCzDppy]₂Cu₄I₄ was negligible. To exclude the influence of host matrix, the temperature-dependent TRES of pure films for the clusters should be compared.

4. The authors reported a very high EQE of 23.5% for a spin-coated white light-emitting device. Because the authors claimed host-cluster energy transfer was limited, the repeatability of the high performances was doubted. Please give the data to demonstrate the performance repeatability of the best devices.

5. Please describe the synthesis details and single crystal growth approach of [tBCzDppy]₂Cu₄I₄ in the supporting information.

Responses to Reviewers' Comments

Reviewer #1:

General Comment: In this paper, Xu et al. designed two copper-iodide clusters of yellow emission, named [Dppy]₂Cu₄I₄ and [tBCzDppy]₂Cu₄I₄. The latter one was doped into a blue-emissive TADF host, named CzAcSF, and fabricated a white-emission CLED device with EQE up to 23.5% at a high doping concentration of 30%. To explain the outstanding performance, photophysical measurement, and quantum chemical calculation were carried out by the authors. The results showed that the tBu-carbazole-modified ligand (tBCzDppy) provides deeper occupied orbitals and high-lying excited states, which enhance the efficiency of charge transfer and energy transfer. These factors lead to the high EQE of the device. The structure-relationship is clear and the molecular design strategy is instructive, so I recommend this paper to be published on Nat. Commun. after addressing the following comments.

Response: We highly appreciate reviewer's pertinent comments and accurate summary of our work! Thanks a lot!

1. Page 1, Line 15, "yellow Cu₄I₄" or "yellow emissive Cu₄I₄"?

Response: Thanks a lot for this kind reminding! We should apologize for our carelessness. As reviewer pointed out, "yellow emissive Cu₄I₄" is correct. All the related expressions were revised accordingly. Thanks a lot!

Revision:

The related expressions were corrected:

"doped by yellow emissive Cu₄I₄ cluster"

"extremely low doping concentrations (< 1%) of yellow emissive dopants"

"one feasible strategy is incorporating yellow emissive dopant"

"since the maximum η_{EQE} of yellow emissive Cu₄I₄ cubes was still less than 10%."

"blue-emitting TADF host and yellow emissive cluster dopant"

2. Page 4, Line 94, "high doing concentration" should be "high doping concentration".

Response: Thanks a lot for this kind reminding! We should apologize for our carelessness, inducing the typos, which were corrected in the revision. We further rechecked the manuscript carefully to get rid of typos and grammatical errors to our best. Thanks a lot!

Revision:

The typos were corrected:

"high doping concentration"

3. Page 12, Line 369 “Copper iodide cluster synthesis”. Please provide the detailed synthesis process utilized to obtain the clusters. Additionally, include characterizations of the Cu_4I_4 clusters, including NMR spectroscopy, mass spectrometry (MS), and elemental analysis either in the main text or in the SI.

Response: Thanks a lot for this constructive comment! The descriptions of preparation procedures for the clusters were enriched in the manuscript. All the details were further described in the experimental section of Supplementary information. All the NMR and MS spectra were added in the supplementary information as Supplementary Figures S35-S40. Thanks a lot!

Revision:

The detailed synthesis process and characterization of clusters include $^1\text{H-NMR}$, $^{13}\text{C-NMR}$, elemental analysis and mass spectrometry (MS) were add.:

“Synthesis of copper iodide clusters. 1 mmol of ligands and 2 mmol of CuI were dissolved in dichloromethane (5 mL) and stirred for four hours at room temperature to afford crude clusters as precipitates, which were recrystallized with dichloromethane and ether to afford pure clusters as yellow crystals. Detailed experimental procedures and structural characterization (Supplementary Figs 35-40) are provided in the Supplementary Information.”

“[Dppy] $_2$ Cu $_4$ I $_4$: In Ar, Dppy (0.26 g, 1 mmol) and CuI (0.38 g, 2 mmol) were dispersed in 5 mL of CH_2Cl_2 . The mixture was stirred for 4 h at room temperature. Then, the solvent was evaporated to obtain crude material, which was further recrystallized from CH_2Cl_2 /ether solution to afford yellowish crystal of 0.58 g with a yield of 90%. ^1H NMR (TMS, CDCl_3 , 400 MHz): δ = 9.043 (d, J = 3.2 Hz, 2H), 7.947 (t, J = 7.6 Hz, 2H), 7.630 (t, J = 5.2 Hz, 2H), 7.438 (t, J = 5.2 Hz, 2H), 7.403 (d, J = 6.8 Hz, 4H), 7.279 ppm (q, J_1 = 6.8 Hz, J_2 = 14 Hz 16H); ^{13}C NMR (TMS, CDCl_3 , 100 MHz): δ = 134.0, 133.8, 132.7, 132.5, 130.9, 129.5, 129.4, 129.0 ppm. ESI-MS: m/z (%) 1287.51 (100) [M^+]; elemental analysis for $\text{C}_{34}\text{H}_{28}\text{Cu}_4\text{I}_4\text{N}_2\text{P}_2$: calculated: C 31.70, H 2.19, N 2.17; found: C 31.73, H 2.22, N 2.19.”

“[tBCzDppy] $_2$ Cu $_4$ I $_4$: In Ar, tBCzDppy (0.54 g, 1 mmol) and CuI (0.38 g, 2 mmol) were dispersed in 5 mL of CH_2Cl_2 . The mixture was stirred for 4 h at room temperature. Then, the solvent was evaporated to obtain crude material, which was further recrystallized from CH_2Cl_2 /ether solution to afford yellow crystal of 1.30 g with a yield of 95%. ^1H NMR (TMS, CDCl_3 , 400 MHz): δ = 9.686 (s, 2H), 7.645 (s, 8H), 7.215 (d, J = 8 Hz, 4H), 6.973 (s, 6H), 6.891 (t, J = 7.2 Hz, 6H), 6.776 (d, J = 6 Hz, 4H), 6.713 (d, J = 6.4 Hz, 8H), 1.346 ppm (s, 36H); ^{13}C NMR (TMS, CDCl_3 , 100 MHz): δ = 143.6, 140.0, 133.6, 133.5, 129.8, 127.6, 127.5, 123.7, 116.0, 109.1, 34.7, 32.1 ppm; ESI-MS: m/z (%) 1842 (100.0) [M^+]; elemental analysis for $\text{C}_{74}\text{H}_{74}\text{Cu}_4\text{I}_4\text{N}_4\text{P}_2$: calculated: C 48.22, H 4.05, N 3.04; found: C 48.25, H 4.08, N 3.07. CCDC No. 2298772.”

Supplementary Fig. 35 | ¹H NMR spectrum of [Dppy]₂Cu₄I₄.

Supplementary Fig. 36 | ¹³C NMR spectrum of [Dppy]₂Cu₄I₄.

Supplementary Fig. 37 | ESI-MS spectrum of $[\text{Dppy}]_2\text{Cu}_4\text{I}_4$.

Supplementary Fig. 38 | ^1H NMR spectrum of $[\text{tBCzDppy}]_2\text{Cu}_4\text{I}_4$.

Supplementary Fig. 39 | ^{13}C NMR spectrum of $[tBCzDppy]_2Cu_4I_4$.

Supplementary Fig. 40 | ESI-MS spectrum of $[tBCzDppy]_2Cu_4I_4$.

4. Supporting information, page 21, line 168 “table S₁”, the energy of S₁ was obtained by the edge of absorption spectra, but the energy of T₁ was obtained by the emission spectra. I’m not sure how to confirm the 0-0 transition energy of the T₁-S₀ state. Please check and revise accordingly.

Response: Thanks a lot for this constructive comment! We should apologize for our unclear description of estimating the T₁ energy levels. Actually, we measured the time-resolved phosphorescence (PH) spectra of the Cu₄I₄ cluster films after a delay of 100 μs to get rid of the interferences from fluorescence (Supplementary Fig. 16). The T₁ energy levels were estimated according to the peak wavelengths of the PH spectra. The T₁ values in Figure 1e and Supplementary Table 1 were updated. Thanks a lot!

Revision:

PH spectra were obtained through slicing time-resolved emission spectra (TRES) in the time ranges of > 100 μ s were add:

“[e] estimated according to peak wavelengths of time-resolved phosphorescence spectra (Supplementary Fig. 11) after a delay of 100 μ s;”

Supplementary Fig. 15 | Time-resolved phosphorescence spectra after 100 μ s of (a) **[Dppy]₂Cu₄I₄** and (b) **[tBCzDppy]₂Cu₄I₄** based spin-coating neat films.

Fig. 1 d Proposed mechanisms of facilitated charge (above) and energy transfer (below) between CzAcSF and [tBCzDppy]₂Cu₄I₄ mediated by tBCz-contributed energy levels, in comparison to [Dppy]₂Cu₄I₄. Data out and in parenthesis are experimental and simulated values, respectively.

5. Supporting Information, page 22, line 177 “table S2”, please provide equations that you used to calculate those rate constants of each photophysics process.

Response: Thanks a lot for this constructive comment! We should apologize for our insufficient description. The relative equations were also added in the Supplementary Note 4. Photophysical Analysis. Thanks a lot!

Revision:

The description and related equations for calculating the transition rate constants were add :

“The calculation formulas for the rate constants of prompt fluorescence (k_{PF}), delayed fluorescence (k_{DF}), singlet radiation (k_r^S), singlet (k_{nr}^S) and triplet nonradiation (k_{nr}^T), reverse intersystem crossing (k_{RISC}) and intersystem crossing (k_{ISC}), and corresponding quantum efficiencies (ϕ) are expressed as following list:

$$k_{PF} = k_r^S + k_{nr}^S + k_{ISC} \quad (\text{Eq. S1})$$

$$k_{DF} = k_{nr}^T + \left(1 - \frac{k_{ISC}}{k_{PF}}\right) \cdot k_{RISC} \quad (\text{Eq. S2})$$

$$k_r^S = \phi_{PF} \cdot k_{PF} \quad (\text{Eq. S3})$$

$$k_{nr}^S = k_{PF} - k_r^S - k_{ISC} = k_{PF} - k_r^S - k_{PF} \cdot \frac{\eta_{PF}}{\eta_{PL}} \quad (\text{Eq. S4})$$

$$k_{ISC} = (1 - \phi_{PF}) \cdot k_{PF} \quad (\text{Eq. S5})$$

$$k_{nr}^T = k_{DF} - \left(1 - \frac{k_{ISC}}{k_{PF}}\right) \cdot k_{RISC} = k_{DF} - \left(1 - \frac{k_{ISC}}{k_{PF}}\right) \cdot \frac{k_{DF} \cdot k_{PF} \cdot \phi_{DF}}{k_{ISC} \cdot \phi_{PF}} \quad (\text{Eq. S6})$$

6. In the main text, please provide further descriptions of the reasons why the authors choose to modify the group on the ortho position of the carbon in the Dppy ligand, as well as the reason for selecting tert-butyl carbazole as the modification.

Response: Thanks a lot for this constructive comment! In our molecular design, a hole-transporting group with host characteristics would provide intermediate energy levels to support energy and charge transfer to coordination core. Carbazole is one of the most popular functional groups widely used in hole transporting materials and host matrixes (TCTA in *Adv. Mater.* **1994**, 6, 677; mCP in *Appl. Phys. Lett.* **2003**, 82, 2422). Considering the T₁ energy level of ~2.2 eV for orange emission from [Dppy]₂Cu₄I₄, the T₁ energy levels of carbazole derivatives are 2.5-2.7 eV, located between those of [Dppy]₂Cu₄I₄ segment and CzAcSF (2.9 eV), therefore, di-tert-butyl carbazole (tBCz) can serve as the self-host group to promote energy transfer from CzAcSF to [Dppy]₂Cu₄I₄. Furthermore, the similar ionization potential of tBCz and carbazole group in CzAcSF, which can facilitate hole transfer from CzAcSF to [tBCzDppy]₂Cu₄I₄.

On the other hand, the *ortho*-substitution of tBCz not only increases steric hindrance, but also induces locally asymmetric configuration of [tBCzDppy]₂Cu₄I₄, which can reduce inter-cluster interaction induced quenching. Carbazole group at the ortho position of Dppy is the closest to the coordination core of Cu₄I₄. Consequently, the distance between tBCz and Cu₄I₄ cube is only 7.5 Å, which is highlighted in Supplementary Fig. 4, and beneficial to energy and charge transfer from the peripheral ligands to the coordination core. Supplementary Fig. 4 was updated, and the detailed explanation was added in the related section. Thanks a lot!

Revision:

Supplementary Fig. 4 and detailed explanation was added:

“Besides the hole-transporting ability, the first triplet (T₁) energy levels of tBCz derivatives (2.5-2.7 eV) are between those of [Dppy]₂Cu₄I₄ (2.3 eV) and CzAcSF (2.9 eV). At ortho-position of DPP, the steric hindrance of tBCz group can effectively reduce inter-cluster interaction induced quenching.”

“Nonetheless, the distance between tBCz and Cu₄I₄ cube is only 7.5 Å, which is beneficial to intra-cluster energy and charge transfer from the peripheral ligands to the coordination core (Supplementary Fig. 4). Therefore, the steric hindrance of tBCz groups benefits quenching suppression and energy and charge transfer, and also improves the solubility of [tBCzDppy]₂Cu₄I₄ in common solvents, e.g. chlorobenzene, making device fabrication by solution processing feasible.”

Supplementary Fig. 4 | Single crystal structure of $[tBCzDppy]_2Cu_4I_4$. The brown dash line indicates the centroid-centroid distances of tBCz and Cu_4I_4 core.

7. Supporting Information, page 21, line 171 “table S1”, the words “[h] in PMMA matrix” but it is not referred to in the table. Please check it carefully.

Response: Thanks a lot for this kind reminding! We should apologize for our carelessness making this mistake. We measured PLQY values of the clusters doped in polymethyl methacrylate (PMMA) films. The corresponding superscripts were corrected. Thanks a lot!

Revision:

The corrected table was added:

Supplementary Table 1. Physical properties of the clusters.

Cluster	$\lambda_{Abs}^{[a]}$ (nm)	$\lambda_{PL}^{[b]}$ (nm)	S_1 (eV)	T_1 (eV)	$\Delta E_{ST}^{[f]}$ (eV)	$\tau^{[g]}$ (μs)	PLQY (%)	HOMO (eV)	LUMO (eV)	FWHM ^[j] (nm)
$[Dppy]_2Cu_4I_4$	275, 347	548	2.33 ^[c] , 2.51 ^[d]	2.28 ^[e] , 2.46 ^[d]	0.05	2.95 (0.68),	1 ^[b] ,	-5.04 ^[d] ,	-1.90 ^[d] ,	148
						8.80 (0.32)	10 ^[h]	-5.98 ^[i]	-3.25 ^[i]	
$[tBCzDppy]_2Cu_4I_4$	297,328, 342	590	2.34 ^[c] , 2.33 ^[d]	2.15 ^[e] , 2.27 ^[d]	0.19	6.66 (0.31),	37 ^[b] , 80 ^[h]	-5.07 ^[d] , -5.71 ^[i]	-2.09 ^[d] , -3.16 ^[i]	147
						6.85				

[h] in polymethyl methacrylate (PMMA) film;

8. Supporting Information, page 22, line 177 “table S2”, please tell the details of how you measure the PLQY of [b] blue and [c] yellow components of CzAcSF:x% cluster films, respectively.

Response: Thanks a lot for this constructive comment! The profiles of white emissions from CzAcSF:x% [tBCzDppy]₂Cu₄I₄ were divided into a blue and a yellow components through double-peak fitting. The contributions of blue and yellow components to emission were estimated according to their corresponding ratios in peak areas, which were further used to calculate their contributions to whole PLQYs, namely nominal blue and yellow PLQY values. Supplementary Fig. 24 and related description were added in the revision. Thanks a lot!

Revision:

The description of blue and yellow PLQY estimation was added:

“which were estimated according to blue and yellow ratios in whole white emission profiles (Supplementary Fig. 25)”

Supplementary Fig. 24 | Double-peak fitting of PL profile for CzAcSF:30% [tBCzDppy]₂Cu₄I₄ film and corresponding blue and yellow peak areas for PLQY estimation.

Reviewer #2:

General Comment: This is an interesting study that presents singly doped white LEDs with EQE beyond 23% at extremely high doping concentrations. Exciton kinetics have been investigated and a relation for the higher efficiency has been fashioned reasonably. However, the discussion is insufficient and not clear enough. In spite of excellent EQE of the white LED reported with exciton studies, the materials reported here and the current state of work is inadequate to be published in this Nature sister journal.

Response: We highly appreciate reviewer's approval on our work! We should apologize for our insufficient discussions and explanations. So, we would like to take this chance to explain the novelty and important findings of this work, especially overcoming the great challenging in efficient energy and charge transfer from TADF hosts to cluster emitters through ligand engineering. Purposeful ligand functionalization can comprehensively optimize optoelectronic properties of cluster based multicomponent systems, which would largely extend the applications of these emerging materials.

Despite the rapid development of cluster-based mono-color light-emitting diodes with external quantum efficiencies more than 20%, only few efficient white cluster light-emitting diodes were reported, mainly due to the inefficient and balanced exciton allocation between blue and yellow emitters in cluster-involved white-emitting systems, especially the uncontrollable energy transfer from blue emitters to clusters. In our previous works, we can not achieve the simultaneous emissions from both clusters and blue-emitting TADF hosts, e.g. CzAcSF, despite carefully tuning cluster concentrations (*Angew. Chem. Int. Ed.* **2023**, *62*, e202305018). Actually, until present, the excited state properties of metal haloid luminescent materials are still not clear. Copper iodide clusters exhibit completely different excited state properties from copper complexes, pure organic molecules and polymers. We believe that this work provides valuable progress in terms of luminescent mechanism. Obviously, the energy transfer and exciton allocation between hosts and cluster emitters is completely different with organic host-guest systems. Therefore, this work not only develops an effective approach to realize white emissions from cluster-based systems, but also demonstrates the feasibility and importance of ligand functionalization in modulating host-cluster energy transfer and exciton allocation in devices. These results can undoubtedly facilitate the development of cluster materials for optoelectronic applications.

The main findings in our work are as follows:

(i) **Ligand engineering modulating excited-state characteristics for yellow Cu₄I₄ clusters.** Different to Cu₄I₄ cubes reported in our previous works, parallelogram copper clusters formed by N[^]P ligands in this work reveal the shortened Cu-Cu and Cu-I distances, resulting in their predominant iodine-ligand (ILCT) and metal-iodine-ligand charge transfer (MXLCT) of the first single and triplet excited states of the clusters, which are different from the intramolecular charge transfer in CzAcSF as blue-emitting TADF host, thus avoiding excessive energy transfer and making high doping concentrations of the cluster emitters.

(ii) **Ligand antenna effect accurately optimizing energy transfer between host matrix and cluster dopants.** We introduce di-(tert-butyl)-carbazole (tBCz) group contributing to the high-lying energy levels and pyridine to form a donor-acceptor structured ligand tBCzDppy, which provides a suitable intermediate energy level between CzAcSF host and cluster core to establish the channels for energy transfer and exciton allocation between CzAcSF and

yellow-emitting cluster core, leading to white emission combining blue and yellow components from CzAcSF and [tBCzDppy]₂Cu₄I₄ at extremely high doping concentration reaching 30%.

(iii) **Excellent EL performance of white light-emitting diodes (LED).** The EQE of the white device based on CzAcSF:30% [tBCzDppy]₂Cu₄I₄ was as high as 23.5%, which was one of the highest values among all kinds of solution-processed white devices reported at present, ensuring the practical value of cluster based white LED.

Therefore, we believe that these results and important findings figure out the feasibility of ligand engineering for excited-state optimization and the importance of ligands in energy transfer processes involved in clusters, which can be referable for subsequent researches. The related discussions were added in the revision. Thanks a lot!

Revision:

The discussions on the important findings were add:

“Therefore, the incorporation of tBCz groups reduces the gap between optoelectronic properties of blue-emitting TADF host and yellow emissive cluster dopant, which is embodied as the antenna effect of tBCzDppy ligands providing the intermediate energy levels to modulate charge and energy transfer processes. These results demonstrate the superiority of cluster emitters in the diversity and tunability of excited-state characteristics, and the importance of ligand engineering for the controllable optimization.”

“These results not only demonstrate the superiorities of cluster emitters for practical white lighting applications, but also indicate the flexible manipulation of excited-state properties for cluster molecules and exciton processes in organic-cluster hybrid systems through ligand engineering.”

1. Researchers did not conduct any studies to further understand the morphological properties of the synthesized materials. Since the devices were fabricated using solution processing technique, morphology is among the crucial factors affecting device performance and stability.

Response: Thanks a lot for this constructive comment! Accordingly, we measured atom force microscopy (AFM) of spin-coated films for neat clusters and CzAcSF:30% clusters with the thickness of 40 nm, which showed uniform and smooth surfaces with root-mean-square (RMS) roughness less than 1 nm. Nonetheless, compared with [Dppy]₂Cu₄I₄ based films, the RMS values of [tBCzDppy]₂Cu₄I₄ based films markedly decrease, indicating the improved film formability and morphological stability of [tBCzDppy]₂Cu₄I₄ by tBCz modification, as well as the enhanced compatibility with CzAcSF as organic matrix. The discussion was added in the relative section, and AFM images were added in Supplementary Fig. 25. Thanks a lot!

Revision:

The Discussion on film morphology and AFM images of spin-coated films were added:

Supplementary Fig. 25 | AFM images of spin-coated films for neat clusters and CzAcSF:30% clusters with the thickness of 40 nm (testing area: $3 \mu\text{m} \times 3 \mu\text{m}$). R_{sq} refers to root-mean-square (RMS) roughness. Compared to $[\text{Dppy}]_2\text{Cu}_4\text{I}_4$ based films, RMS value of neat $[\text{tBCzDppy}]_2\text{Cu}_4\text{I}_4$ film is nearly halved, and RMS value of CzAcSF: $[\text{tBCzDppy}]_2\text{Cu}_4\text{I}_4$ is also reduced by one quarter, which demonstrate tBCz modification improves film formability, morphological stability and compatibility with organic host matrixes of $[\text{tBCzDppy}]_2\text{Cu}_4\text{I}_4$.

“Compared to $[\text{Dppy}]_2\text{Cu}_4\text{I}_4$, tBCz modification improves the film formability, morphological stability and matrix compatibility of $[\text{tBCzDppy}]_2\text{Cu}_4\text{I}_4$, making device fabrication by solution processing feasible (Supplementary Fig. 25).”

2. The researchers did not investigate the charge transport properties of the cluster materials and doped TADF films.

Response: Thanks a lot for this constructive comment! Accordingly, we fabricated nominal single-carrier-transporting devices based on layers of neat clusters and CzAcSF:30% clusters. It shows that compared to $[\text{Dppy}]_2\text{Cu}_4\text{I}_4$, the intrinsic electron and hole transporting abilities of $[\text{tBCzDppy}]_2\text{Cu}_4\text{I}_4$ were simultaneously improved by 1-2 orders of magnitude, owing to enhanced intra-ligand donor-acceptor interactions between tBCz and pyridine. While, carrier transporting abilities of CzAcSF matrix were stronger than the clusters, therefore, the differences between current

densities of the cluster doped devices were reduced. Nonetheless, the hole transporting ability of [tBCzDppy]₂Cu₄I₄ doped film was still stronger than that of [Dppy]₂Cu₄I₄ doped analog, manifesting the contribution of tBCz groups in hole transportation and their predominance in hole injection into [tBCzDppy]₂Cu₄I₄. The discussion on electrical properties of the clusters and Supplementary Fig. 34 of IV characteristics for single-carrier transporting devices were added in the revision. Thanks a lot!

Revision:

Discussion on carrier transporting abilities of the clusters was add:

Supplementary Fig. 34 | *IV* characteristics of single-carrier-transporting devices based on [Dppy]₂Cu₄I₄ and [tBCzDppy]₂Cu₄I₄. (a) Voltage-current density (J) curves of single-carrier-transporting devices with configurations of ITO|PEDOT:PSS (40 nm)| [Dppy]₂Cu₄I₄ or [tBCzDppy]₂Cu₄I₄ (40 nm)|MoO₃ (6 nm)|Al (100 nm) for hole-only (shallow symbols) and ITO|LiF (1 nm)| [Dppy]₂Cu₄I₄ or [tBCzDppy]₂Cu₄I₄ (40 nm)|LiF (1 nm)|Al (100 nm) for

electron-only (solid symbols), respectively; (b) Voltage-current density (J) curves of single-carrier-transporting devices for ITO|PEDOT:PSS (40 nm)|CzAcSF: $x\%$ [Dppy]₂Cu₄I₄ or [tBCzDppy]₂Cu₄I₄ (40 nm)|MoO₃ (6 nm)|Al (100 nm) and ITO|LiF (1 nm)|CzAcSF: $x\%$ [Dppy]₂Cu₄I₄ or [tBCzDppy]₂Cu₄I₄ (40 nm)|LiF (1 nm)|Al (100 nm), respectively. $x = 0$ for neat CzAcSF, and $x = 30\%$.

“The IV characteristics of nominal single-carrier transporting devices show that that the carrier transporting abilities of [tBCzDppy]₂Cu₄I₄ are 1-2 orders of magnitude stronger than those of [Dppy]₂Cu₄I₄ (Supplementary Fig. 33). Despite the predominance of CzAcSF in electrical process, hole-only current densities of CzAcSF:30% [tBCzDppy]₂Cu₄I₄ based devices were still larger than those of CzAcSF:30% [Dppy]₂Cu₄I₄ based analogues. It manifests tBCz groups establish hole injection and transportation channels from CzAcSF matrix to the cluster dopants.”

3. No device lifetime (T50 and T90) of the fabricated devices were discussed in the manuscript.

Response: Thanks a lot for this constructive comment! Actually, for solution-processed devices, the optimization of carrier injection and transportation are difficult, due to their simple structures with limited carrier transporting layers. Especially for hole transportation, spin-coated devices commonly only used PEDOT:PSS for hole injection, but did not use hole transporting layers, because spin-coating emissive layers on hole transporting layers can easily destroy the latter and make two layers mixed, leading to electroluminescent performance decrease. Therefore, optimizing the lifetime of solution-processed devices is considerably more time consuming than for vacuum processes because variation of charge transport layers requires the development of cross-linkable materials and careful control of layer morphology. A common strategy is to use vacuum-processable model emitters and realize the resulting, optimized stack with cross-linkable analogues (*Angew. Chem.* **2013**, 52, 9563). Therefore, most of CuI based spin-coated devices did not report device duration data (*Adv. Mater.* **2015**, 27, 2538). In our work, since no suitable cross-linkable hole-transporting materials were commercially available, we also adopted the simple device structures without hole-transporting layers. In this case, the long duration time can be hardly achieved.

Nonetheless, we believe that the triplet exciton allocation to [tBCzDppy]₂Cu₄I₄ can effectively alleviate exciton accumulation on CzAcSF, therefore elongate device duration. To demonstrate it, we performed aging experiments of the spin-coated devices based on CzAcSF:30% [tBCzDppy]₂Cu₄I₄ (Figure C1). The control devices based on neat CzAcSF as emissive layers were also fabricated and measured for comparison. It showed that with the initial luminance of 2000 cd m⁻², the duration time at ~1000 cd m⁻² was 0.052 hours for CzAcSF:30% [tBCzDppy]₂Cu₄I₄, which was about seven folds of that of neat CzAcSF based analogs. Thus, it is rational that the incorporation of [tBCzDppy]₂Cu₄I₄ into CzAcSF layer indeed retarded the device aging. The short device duration of the white-emitting devices should be the combined result of the drawback for TADF-featured CzAcSF and the unbalanced carrier fluxes under the simple spin-coated structures. The description was added in the relative section. Thanks a lot!

Figure C1. Comparison on durations of CzAcSF and CzAcSF:30% [tBCzDppy]₂Cu₄I₄ based devices. The measurement was performed under constant current mode with initial luminance of 2000 cd m⁻².

Revision:

Device duration was discussed:

“Although TADF feature of CzAcSF and simple structures of the spin-coated devices could reduce duration times, the efficiency roll off was only 12.7% at 100 cd m⁻², which was outstanding among solution-processed devices, reflecting the modified carrier recombination and suppressed exciton quenching.”

4. No device stability (in different conditions) and reproducibility of the fabricated devices were discussed in the manuscript.

Response: Thanks a lot for this constructive comment! The device repeatability and performance stability of CzAcSF:30% [tBCzDppy]₂Cu₄I₄ based white-emitting diodes were verified through statistical analysis on the maximum EQE values of 20 devices with the same structures. The average value of the maximum EQE was 23.5%, accompanied by the maximum and minimum values of 25.5% and 21.5%. The distribution of the maximum EQE values was normal, with a small relative standard deviation less than 5%. The discussion and Supplementary Fig. 30 were added in the revision. Thanks a lot!

Revision:

The histogram of peak EOE was added::

Supplementary Fig. 30 | Statistic analysis of the repeatability for the maximum EQE values of 20 CzAcSF:30% [tBCzDppy]₂Cu₄I₄ devices with the same structures. RSD refers relative standard deviation. The EQE variation followed normal distribution with a small RSD less than 5%.

“The repeatability of CzAcSF:30% [tBCzDppy]₂Cu₄I₄ based devices were verified by their stable efficiencies (Supplementary Fig. 30).”

5. The solvent(s) used for device fabrication were not stated in the manuscript.

Response: Thanks a lot for this mind reminding! We should apologize for our carelessness during manuscript preparation. The clusters and hosts were dissolved in chlorobenzene with the total concentrations of 10 mg ml⁻¹. This information was added in “Device fabrication” section of supplementary materials. Thanks a lot!

Revision:

Device fabrication condition was revised:

“CzAcSF, BCPO and clusters were dissolved in chlorobenzene with the total concentrations of 10 mg ml⁻¹.”

6. There is no discussion on the turn-on voltage (V_{on}) differences observed in different devices. In addition, in the table S3 and S4, the voltages measured for 1 cd/m² were displayed. However, in figures S17-22, no intercept was observed at 1 cd/m² which is very bizarre. In Figure 4c and S18a, voltages used for devices measurements were random and are not in order. This is the scenario in other device JVL graphs but it is too prominent in these two graphs

except Figure S20. Figure S20 itself is very erratic and not following any trend. Even so, no explanation for such uncommon performance observed was discussed.

Response: Thanks a lot for this constructive comment! As summarized in Supplementary Tables 3 and 4, compared to neat CzAcSF based devices, cluster doping markedly increased driving voltages, indicating the dominance of CzAcSF matrix in carrier injection and transportation. Furthermore, compared to [Dppy]₂Cu₄I₄ based analogues, no matter CzACsF or BCPO used as hosts, [tBCzDppy]₂Cu₄I₄ based devices revealed the lower driving voltages and higher luminance, evidencing the involvement of [tBCzDppy]₂Cu₄I₄ in electrical processes and improved exciton formation and radiation via tBCz modification.

We should apologize for the confusing expressions. As reviewer pointed out, at turn-on voltages, the luminance of some devices was more than 1 cd m⁻². Therefore, the footnotes of Supplementary Tables 3 and 4 about “voltages at 1 cd m⁻²” were incorrect, which was revised as “operation voltages for turn on”. To avoid operation errors, the devices were firstly measured from 0 V to determine the turn-on voltages, and then formally remeasured from the turn-on voltages with the same voltage intervals. Since the turn-on voltages of the devices were different, the voltage variations of the devices were also diverse.

In Supplementary Fig. 32, since the luminance of BCPO:x% [Dppy]₂Cu₄I₄ based devices was very low (about 30 cd m⁻² for the maximum), the measurement results were sensitive to operation and unstable. Therefore, the data of BCPO:x% [Dppy]₂Cu₄I₄ based devices seemed erratic. Nonetheless, the luminance and efficiencies of the devices still indicated the tendencies of first increase and then decrease. The turning point was at $x = 20$.

The discussion on operation voltages was added, and footnotes of Supplementary Tables 3 and 4 and Supplementary Fig. 32 were revised. Thanks a lot!

Revision:

Discussion on driving voltages was added, and Supplementary Fig. 30 and tables 3 and 4 were revised:

“Compared to [Dppy]₂Cu₄I₄ based analogues, [tBCzDppy]₂Cu₄I₄ endowed its devices with the lower driving voltages and higher luminance (Supplementary Tables 3 and 4). Therefore, tBCz modification made [tBCzDppy]₂Cu₄I₄ involved in electrical processes, and improved exciton formation and radiation.”

“The devices were firstly applied with bias from zero to find out the turn-on voltages, and then formally measured from the turn-on voltages to minimize operation errors.”

“[a] Operation voltages for turn on, and at 100 and 1000 cd m⁻²”

Supplementary Fig. 32 | (a) EL spectra (inset) and Current density (J)-Voltage-Luminance characteristics of BCPO:*x*% [Dppy]₂Cu₄I₄ based CLEDs in doping concentration *x*% range of 10%-40%; (b) Efficiencies vs. Luminance relationships. Despite low luminance and efficiencies, the tendencies of the luminance and efficiencies for the devices were “first increase and then decrease”, corresponding to a turning point at *x* = 20.

7. Overall, despite an outstanding performance observed, the explanation for such performance was not supported properly. Just using DFT results from Gaussian09 and exciton kinetics, the authors tried explaining the improved performance of the devices. A lot of effort has been put to reason the high EQE but even that falls short and rest was either ignored or skipped. The device performance is unorganized and not reliable.

Response: Thanks a lot for this constructive comment! According to reviewer’s suggestions, we performed comprehensive investigations on the influences of tBCz groups on optoelectronic properties of the clusters:

(i) AFM results show that [tBCzDppy]₂Cu₄I₄ based spin coated films achieve the reduced surface roughness, reflecting the improved film formability of [tBCzDppy]₂Cu₄I₄.

(ii) [tBCzDppy]₂Cu₄I₄ has stronger carrier transporting abilities, in comparison to [Dppy]₂Cu₄I₄, and its tBCz groups are involved in carrier transportation to cluster coordination cores, which enhances carrier recombination and exciton allocation to [tBCzDppy]₂Cu₄I₄, and simultaneously alleviates the exciton concentration quenching in CzAcSF matrix.

(iii) tBCz groups provide intermediate energy levels to facilitate host-cluster energy transfer, leading to the high-quality white emission, and especially well-controlled high doping concentrations of clusters, which not only improve EQE values, but also indicate the practical superiorities of cluster luminescent materials.

Moreover, to demonstrate the crucial effect of tBCz groups in providing intermediate energy levels for modulating host-cluster energy transfer, we re-measured time-resolved emission spectra (TRES) of CzAcSF:30% [tBCzDppy]₂Cu₄I₄ with higher time resolution to discover the direct evidence indicating the role of tBCz for energy transfer and exciton allocation. It shows that CzAcSF as the host matrix is firstly excited, then a new emission band occurs at 12-14 μs, corresponding to high-lying triplet state on tBCz, and finally tBCz groups transfer the excited-state energy to the S₁/T₁ states of [tBCzDppy]₂Cu₄I₄. This variation is consistent with DFT simulation results. The discussions on detailed energy transfer process and the effect of ligand engineering on improving device efficiencies, and Supplementary Fig. 23 were added. Thanks a lot!

Revision:

A supplementary discussion of high device performance is added to the text:

Supplementary Fig. 23 | Sliced time resolved emission spectra (TRES) of CzAcSF:30% [tBCzDppy]₂Cu₄I₄ film at room temperature in the time range of 0-55 μs.

“Actually, NTO results show that compared to the S₉ and S₁₀ states with significant contributions from Cu₄I₄, the T₉ and T₁₀ states of [tBCzDppy]₂Cu₄I₄ are typical ³ILCT, whose “holes” and “electrons” are respectively localized on

tBCz donors and pyridine acceptors, which are highly similar to intramolecular charge transfer of CzAcSF. Consistently, sliced TRES of CzAcSF:30% [tBCzDppy]₂Cu₄I₄ indicates a transition band occurring in the time range of 12-40 μ s, corresponding to the intermediate triplet states located on tBCz groups (Supplementary Fig. 22). Thus, tBCzDppy serves as the antenna to facilitate energy transfer in CzAcSF:x% [tBCzDppy]₂Cu₄I₄, through providing its T₉ and T₁₀ states as intermediate states.”

“tBCz modification simultaneously improves the morphological and electrical properties and exciton utilization for [tBCzDppy]₂Cu₄I₄, therefore facilitating host-cluster energy transfer, enhancing device repeatability and alleviating exciton quenching. It is convincing that the unique excited-state characteristics of [tBCzDppy]₂Cu₄I₄ is well complementary with TADF feature of CzAcSF, leading to the desired heavily and singly doped white light-emitting devices with the state-of-the-art efficiencies.”

Reviewer #3:

General Comment: In this work, the authors selected yellow-emitting copper clusters and a blue TADF host CzAcSF to form the white-emitting devices. Although there were reports of white OLEDs and PeLEDs, I agree with the authors that clusters based on eco-friendly cheap metals can provide the important alternatives for the large-scale applications, since the high rigidity of clusters can limit the J-T distortion of mononuclear complexes. I noticed that Yao et al. recently reported a warm white-emitting device of Cu₂I₂ clusters with broad emissions, which was obviously different to this work regarding to material design and photophysical investigation (*Nat Photon* **18**, 200-206 (2024)). The performance reported in this work was also improved. What I am interested in is the heavy doping for realizing white emission reported in this work, which was thoroughly different to all previously reported white-emitting systems. This finding may discover one advantage of cluster emitters. The authors organized the experiments and investigations carefully to figure out the fundamentals behind the unique photophysical processes in CzAcSF:Cu₄I₄ cluster films and the key factors of carrier and energy transfer. The EQE of 23.5% achieved at 30% doping concentration by the solution-processed devices indicated the potential of white cluster light-emitting devices. Therefore, I would like to recommend the publication of this work in Nature Communications.

Response: We highly appreciate reviewer's pertinent comments and summary of our work! Thanks a lot!

1. It was a good idea to use copper clusters as long-wavelength emitters for binary white-emitting systems, considering the sustainability and low cost of copper, since previous works were based on noble-metal complexes containing iridium and platinum, etc. For example, Forrest et al. reported the white OLEDs based on Ir(ppy)₃ and PQIr (*Nature* **440**, 908-912 (2006)). But, I still wonder are there any other fundamental advantages of the copper clusters for white-emitting applications.

Response: Thanks a lot for this constructive comment! Actually, single-emissive-layer white light-emitting diodes attract much attention, since this structure can effectively improve performance repeatability and device stability through minimizing layer numbers and interfaces, among which singly doped single-emissive-layer structures would be "ideal". However, for noble-metal complexes based systems, the doping concentrations of long-wavelength phosphors should be very low to avert excessive energy transfer, markedly reducing the stabilities of color purity, efficiencies and duration for the corresponding white devices. In this case, a feasible strategy is to develop yellow dopants with similar but different excited state properties to the blue-emitting host matrix. According to our previous works, different to mononuclear copper complexes, luminescent copper clusters show unique multi-component excited states due to their more complicated intra-ligand, ligand-metal and metal-metal interactions, in which ligand centered excited states are similar to intramolecular charge transfer excited states of TADF hosts, but cluster-involved excited states are different. In this case, ligand encapsulation and different composition of cluster emitters can be tuned to control the host-guest energy transfer, making heavy doping based white emission feasible. In this work, we demonstrate that through ligand engineering, the ligand based high-lying excited states can serve as the intermediate energy levels to facilitate the energy and carrier transfer from CzAcSF matrix. As consequence, the state-of-the-art

efficiencies and high white color purities were achieved by [tBCzDppy]₂Cu₄I₄, based on not only singly doped emissive layer structure, but also cluster concentrations as high as 30%. The explanation on superiorities of cluster emitters for white device applications was added in the revision. Thanks a lot!

Revision:

The detailed explanation on energy transfer optimization through cluster emitters was added:

“Ligand-centered (LC) components, especially intra-ligand charge transfer states, are similar to those of phosphors and TADF molecules, but the CC state is completely different. The charge and energy transfer from blue-emitting host matrixes can be facilitated by the former, but limited by the latter, leading to exciton allocation balance. In this sense, through optimizing LC and CC characteristics and ratios in excited states, luminescent clusters would be competent as yellow/orange emitters in singly and heavily doped white light-emitting systems.”

2. On page 7, paragraph 3, the authors claimed that the emission duration of [Dppy]₂Cu₄I₄ film decreased more significantly at higher temperatures, in comparison to [tBCzDppy]₂Cu₄I₄ film, because the rigid and bulky tBCz groups inhibited the quenching effect. Since tBCz groups also modified the solubility and film formability, the film morphology of [tBCzDppy]₂Cu₄I₄ might be different with that of [Dppy]₂Cu₄I₄, the authors are suggested to add and discuss the time decay curves of the clusters in powder.

Response: Thanks a lot for this constructive suggestion! Accordingly, we measured the time decay curves of the clusters in powder (Supplementary Fig. 13). Compared with [Dppy]₂Cu₄I₄ powder, the variation of emission time decays for [tBCzDppy]₂Cu₄I₄ powder is still smaller, along with temperature increasing, which further verifies the effect of tBCz modification on quenching suppression. The figure and related discussion were added in the revision. Thanks a lot!

Revision:

Time decays of cluster powders and related discussion were added:

“On the contrary, [tBCzDppy]₂Cu₄I₄ film shows the less temperature-dependent singly exponential time decays, corresponding to lifetimes ~1 μs shorter than those of [Dppy]₂Cu₄I₄ film (Supplementary Table 1), which are consistent with the situation of cluster powders (Supplementary Fig. 13).”

Supplementary Fig. 13 | Time decays of **[Dppy]₂Cu₄I₄** and **[tBCzDppy]₂Cu₄I₄** powders in the range of 20-300 K with an interval of 10 K.

3. On page 8, paragraph 3, CzAcSF:*x*% [tBCzDppy]₂Cu₄I₄ film exhibited dual-peak emission, and the relative intensity of yellow was proportional to *x*%, resulting in emission color change from cool white to warm white. Since the authors claimed the importance and limitation of host-cluster energy transfer, the dependence of emission color on the smaller *x*% change would reflect the improved energy transfer from CzAcSF to [tBCzDppy]₂Cu₄I₄.

Response: Thanks a lot for this constructive comment! Accordingly, we further measured the PL spectra of CzAcSF:*x*% [tBCzDppy]₂Cu₄I₄ with (Supplementary Fig. 17). It shows that the blue/yellow ratio of PL spectra is in reverse proportion to *x*%. The tendency of gradual variation further demonstrates the well-controlled host-cluster energy transfer, owing to the balance of ligand-centered and cluster-centered excited states in energy transfer process. The figure and related discussion were added in the revision. Thanks a lot!

Revision:

The more detailed discussion on the concentration dependence of PL spectra for CzAcSF:*x*% [tBCzDppy]₂Cu₄I₄ films was added:

“Obviously, compared to [Dppy]₂Cu₄I₄, incorporation of tBCz moieties in [tBCzDppy]₂Cu₄I₄ establishes the effective host-cluster energy transfer channel, and makes the accurate modulation of white emissions feasible (Supplementary Fig. 17).”

Supplementary Fig. 17 | PL spectra of CzAcSF:*x*% [tBCzDppy]₂Cu₄I₄ films in the range of *x* = 25-35 with an interval of 1%.

4. The authors claimed that different to Dppy, tBCzDppy may have the TADF characteristics due to its donor-acceptor structure. I suggest them to compare the photophysical properties of Dppy and tBCzDppy to support it, especially temperature-related decays.

Response: Thanks a lot for this constructive suggestion! We measured the temperature-correlated time decay curves of tBCzDppy film. Two components respectively at microsecond and nanosecond scales correspond to delayed and prompt fluorescence from tBCzDppy, indicating its TADF-featured emission. The time decays of tBCzDppy film are nearly temperature independent, reflecting the balance of reverse intersystem crossing and triplet quenching. In contrast, Dppy does not have any microsecond-scale emission. The discussion and Supplementary Fig. 16 were added in the revision. Thanks a lot!

Revision:

Discussion on time decays of spin-coated films based on neat tBCzDppy was added:

“In comparison, tBCzDppy exhibits TADF characteristics similar to CzAcSF, indicating their compatible intermolecular charge transfer states (Supplementary Fig. 16).”

Supplementary Fig. 16 | Time decays of spin-coated film based on neat tBCzDppy (a) at microsecond and nanosecond (inset) scale at room temperature, and (b) in the range of 20-300 K with an interval of 10 K.

5. On page 10, line 305, “At $x=30$, color rendering index (CRI) can reach to 81” was different to the CRI value of 72 in Table S3. The authors should carefully check the data.

Response: Thanks a lot for this kind reminding! We should apologize for our carelessness in manuscript preparation.

“At $x=30$, color rendering index (CRI) can reach to 81” was correct. The wrong number in Supplementary Table 3 was corrected. Thanks a lot!

Revision:

Supplementary Table 3 was revised:

Supplementary Table 3. EL performance of CzAcSF: $x\%$ clusters based devices.

	x (wt%)	$V^{[a]}$ (V)	$L_{\max}^{[b]}$ (cd m ⁻²)	$\eta^{[c]}$			λ_{EL} (nm) / CIE (x, y) ^[d]	CCT ^[e] (K)	CRI ^[f] (Ra)
				η_{CE} (cd A ⁻¹)	η_{PE} (lm W ⁻¹)	η_{EQE} (%)			
CzAcSF: $x\%$ [tBCzDppy] ₂ Cu ₄ I ₄	10	3.6, 5.2, 8.2	1769	32.1, 30.0, 18.6	26.2, 19.1, 8.4	13.6, 12.8, 7.8	492/(0.23, 0.37)	-	-
	20	3.6, 5.3, 8.5	1145	41.4, 35.4, 10.6	35.1, 20.9, 4.0	17.5, 14.4, 4.4	484,555/(0.28, 0.37)	7632	72
	30	3.6, 5.0, 7.3	1169	58.7, 51.3, 27.1	51.2, 31.1, 10.5	23.5, 20.5, 10.8	475,544/(0.33, 0.41)	5675	81
	40	3.7, 6.0, -	386	40.6, 32.3, -	33.7, 16.9, -	15.6, 12.4, -	472,556/(0.33, 0.46)	4532	68

[a] Operation voltages at 1, 100 and 1000 cd m⁻²; [b] the maximum luminance; [c] EL efficiencies at the maximum, 100 and 1000 cd m⁻²; [d] EL peak wavelengths and CIE coordinates at 1000 cd m⁻²; [e] correlated color temperature; [f] color render index.

Reviewer #4:

General Comment: This work reported a novel single-doped white light emission system based on blue TADF host (CzAcSF) and yellow Cu₄I₄ cluster dopant ([tBCzDppy]₂Cu₄I₄). The EL emission changed from cold white to warm white through tuning doping concentration of 20-40%, owing to the limited host-cluster energy transfer. When the doping concentration was 30%, the EQE of the solution-processed devices reached 23.5%. The advantage of singly cluster doped white-emitting device in heavy doping is important for applications. The authors made a systematic investigation and rational explanation on the improved energy transfer between cluster and CzAcSF, through high-lying excited states from carbazole moieties. The authors demonstrated the main conclusions with solid evidences, which would be important to understand the unique excited-state properties of cluster emitters. This work can be published in Nature Communications after suitable revisions.

Response: Reviewer's kind approval and accurate summary of our work is highly appreciated. Thanks a lot!

1. I have noticed that in recent years, there have been some works used clusters as emissive layers to prepared electroluminescent devices, most of which showed blue or yellow emissions, and the reports of white devices based on clusters were only few (Nat. Commun. 2023, 14, 2901; Angew. Chem. Int. Ed. 2021, 61, e202213826). Please briefly introduce the main challenge of developing cluster based white light-emitting devices.

Response: Thanks a lot for this constructive comment! As reviewer pointed out, luminescent cluster materials for monochrome light-emitting devices developed rapidly in recent years, but clusters based white devices were only few. In our previous works, we found that doping yellow-emitting clusters into blue-emitting hosts resulted in either blue or yellow emissions, rather than white emissions, which was even independent on cluster concentrations (Nat. Commun. 2023, 14, 2901). The key challenge is that their different excited-state characteristics hinder the host-cluster energy transfer. Compared with pure organic molecules or mononuclear complexes, clusters with multinuclear metallic cores undoubtedly incorporate much more complicated excited states. Nonetheless, it can be noted that multi-component excited states of clusters include intra-ligand, ligand-metal and metal-metal charge transfer states, in which ligand centered excited states are similar to intramolecular charge transfer excited states of TADF hosts, but cluster-involved excited states are different. In this case, ligand centered excited states can serve as the intermediate states to facilitate energy transfer from TADF hosts to luminescent cluster-involved excited states. This should be based on delicate ligand engineering for optimizing excited-state composition and ratio of the cluster emitters. In this work, through ligand engineering, tBCzDppy based

high-lying excited states serve as the intermediate energy levels to facilitate the energy and carrier transfer from CzAcSF matrix. As consequence, the state-of-the-art efficiencies and high white color purities were achieved by [tBCzDppy]₂Cu₄I₄. The description of main challenge in developing cluster based white light-emitting systems was added in the revision. Thanks a lot!

Revision:

Description of main challenge in developing white light-emitting systems was added:

“Obviously, the key challenge for “ligand-mediated” strategy is how to realize stronger cluster-centered interactions and appropriate ligand functionalization for rationally optimizing excited-state characteristics of the clusters.”

2. Dppy is a common ligand for cluster preparation. Wang et al prepared gold clusters based on Dppy (J. Am. Chem. Soc. 2013, 135, 16184). Therefore, are there any advantages of this kind of ligands in comparison to triphenylphosphine, regarding to EL cluster design.

Response: Thanks for this constructive comment! As a bidentate ligand, Dppy can form stable coordination bonds with nearly all kinds of transition metals through its N and P atoms. Furthermore, N-P distance of Dppy unit is ~3 Å. The limited coordination space reduces inter-copper distances, and enhances Cu-Cu interactions and ligand encapsulation of metallic cores. On the other hand, pyridine with electron-withdrawing effect would contribute to electron injection into the clusters; while, introducing an electron-donating tBCz group forms donor-acceptor (D-A) structure for tBCzDppy with enhanced intra-ligand charge transfer interactions and TADF characteristics, therefore providing energy and charge transfer channels of exciton allocation balance for white emission from host-cluster systems.

Thanks a lot!

Revisions:

The advantages of Dppy over triphenylphosphine were discussed:

“With the purpose to enhance Cu-Cu and Cu-I interactions, Dppy unit as planar and rigid pyridine *ortho*-substituted with flexible diphenylphosphine (DPP) was chosen to form compact bidentate mode, which can form stable coordination with transition metals.”

“Besides the hole-transporting ability, the first triplet (T_1) energy levels of tBCz derivatives (2.5-2.7 eV) are between those of [Dppy]₂Cu₄I₄ (2.3 eV) and CzAcSF (2.9 eV). At ortho-position of DPP, the steric hindrance of tBCz group can effectively reduce inter-cluster interaction induced quenching.”

3. In Fig 2 (d), the time-resolved emission spectra (TRES) of PMMA:10% cluster films revealed that the effect of phonon relaxation on [tBCzDppy]₂Cu₄I₄ was negligible. To exclude the influence of host matrix, the temperature-dependent TRES of pure films for the clusters should be compared.

Response: Thanks a lot for this constructive comment! Accordingly, we measured TRES of neat cluster films at 20-300 K. TRES contours of [Dppy]₂Cu₄I₄ film are markedly shortened along with temperature increasing, due to excited-state structural relaxation. In contrast, the temperature dependence of [tBCzDppy]₂Cu₄I₄ film is much smaller, owing to the suppressed phonon relaxation, which is identical to the situation of PMMA:10% cluster films. TRES data were added as Supplementary Fig. 14 Thanks a lot!

Revisions:

TRES spectra of neat cluster films of clusters and discussions were added:

Supplementary Fig. 14 | TRES spectra of spin-coated neat films for (a) [Dppy]₂Cu₄I₄ and (b) [tBCzDppy]₂Cu₄I₄ at different temperatures.

“Time-resolved emission spectra (TRES) of neat and PMMA:10% cluster films reveal the unchanged transition processes during radiation (Fig. 2d and Supplementary Fig. 14).”

“In contrast, time decays of PMMA:10% [tBCzDppy]₂Cu₄I₄ are markedly elongated at 50-200 K, but comparable to neat film at 300 K (Supplementary Fig. 14).”

4. The authors reported a very high EQE of 23.5% for a spin-coated white light-emitting device. Because the authors claimed host-cluster energy transfer was limited, the repeatability of the high performances was doubted. Please give the data to demonstrate the performance repeatability of the best devices.

Response: Thanks a lot for this constructive comment! The device repeatability and performance stability of CzAcSF:30% [tBCzDppy]₂Cu₄I₄ based white-emitting diodes were verified through statistical analysis on the maximum EQE values of 20 devices with the same structures. The average value of the maximum EQE was 23.5%, accompanied by the maximum and minimum values of 25.5% and 21.5%. The distribution of the maximum EQE values was normal, with a small relative standard deviation less than 5%. The discussion and Supplementary Fig. 30 were added in the revision. Thanks a lot!

Revision:

The histogram of peak EQEs was added::

Supplementary Fig. 30 | Statistic analysis of the repeatability for the maximum EQE values of 20 CzAcSF:30% [tBCzDppy]₂Cu₄I₄ devices with the same structures. RSD refers relative standard deviation. The EQE variation followed normal distribution with a small RSD less than 5%.

“The repeatability of CzAcSF:30% [tBCzDppy]₂Cu₄I₄ based devices were verified by their stable efficiencies (Supplementary Fig. 30).”

5. Please describe the synthesis details and single crystal growth approach of [tBCzDppy]₂Cu₄I₄ in the supporting information.

Response: Thanks a lot for this constructive comment! Accordingly, the experimental details of cluster synthesis and single crystal growth were added in the supplementary materials. Thanks a lot!

Revision:

The details of synthesis procedures and single crystal growth were add.:

[tBCzDppy]₂Cu₄I₄: In Ar, tBCzDppy (0.54 g, 1 mmol) and CuI (0.38 g, 2 mmol) were dispersed in 5 mL of CH₂Cl₂. The mixture was stirred for 4 h at room temperature. Then, the solvent was evaporated to obtain crude material, which was further recrystallized from CH₂Cl₂/ether solution to afford yellow crystal of 1.30 g with a yield of 95%. ¹H NMR (TMS, CDCl₃, 400 MHz): δ = 9.766 (s, 2H), 7.731 (s, 8H), 7.301 (d, *J* = 8 Hz, 4H), 7.053-6.977 (m, 12H), 6.865 (d, *J* = 4.8 Hz, 4H), 6.798 (d, *J* = 6.4 Hz, 8H), 1.433 ppm (s, 36H); ¹³C NMR (TMS, CDCl₃, 100 MHz): δ = 143.6, 140.0, 133.6, 129.8, 127.6, 123.7, 116.0, 109.1, 34.7, 32.1 ppm; LDI-TOF: *m/z* (%) 1842 (100.0) [M⁺]; elemental analysis for C₇₄H₇₄Cu₄I₄N₄P₂: calculated: C 48.22, H 4.05, N 3.04; found: C 48.25, H 4.08, N 3.07. CCDC No. 2298772.

“The crystals suitable for single-crystal XRD analysis were obtained through slowly diffusing ether into dichloromethane solution of the clusters at room temperature.”

REVIEWERS' COMMENTS

Reviewer #1 (Remarks to the Author):

The authors has addressed all the concerns. Now the publication is recommended.

Reviewer #2 (Remarks to the Author):

The authors revised the manuscript according to the reviewers' comments, and its quality is improved.

I think the revised manuscript is acceptable for publication in this journal.

Reviewer #3 (Remarks to the Author):

Authors satisfactorily addressed Referee's comments, the manuscript can be recommended for publication in its current form.

Reviewer #4 (Remarks to the Author):

Since the issues raised have been well addressed, this work could be published.